# Conservative State Value Estimation for Offline Reinforcement Learning

**Liting Chen**[*]
McGill University
Montreal, Canada
98chenliting@gmail.com

**Jie Yan** †
Microsoft
Beijing, China
dasistyanjie@gmail.com

**Zhengdao Shao**[*]
University of Sci. and Tech. of China
Hefei, China
zhengdaoshao@mail.ustc.edu.cn

**Lu Wang**
Microsoft
Beijing, China
wlu@microsoft.com

**Qingwei Lin**
Microsoft
Beijing, China
qlin@microsoft.com

**Saravan Rajmohan**
Microsoft 365
Seattle, USA
saravar@microsoft.com

**Thomas Moscibroda**
Microsoft
Redmond, USA
moscitho@microsoft.com

**Dongmei Zhang**
Microsoft
Beijing, China
dongmeiz@microsoft.com

## Abstract

Offline reinforcement learning faces a significant challenge of value over-estimation due to the distributional drift between the dataset and the current learned policy, leading to learning failure in practice. The common approach is to incorporate a penalty term to reward or value estimation in the Bellman iterations. Meanwhile, to avoid extrapolation on out-of-distribution (OOD) states and actions, existing methods focus on conservative Q-function estimation. In this paper, we propose Conservative State Value Estimation (CSVE), a new approach that learns conservative V-function via directly imposing penalty on OOD states. Compared to prior work, CSVE allows more effective state value estimation with conservative guarantees and further better policy optimization. Further, we apply CSVE and develop a practical actor-critic algorithm in which the critic does the conservative value estimation by additionally sampling and penalizing the states *around* the dataset, and the actor applies advantage weighted updates extended with state exploration to improve the policy. We evaluate in classic continual control tasks of D4RL, showing that our method performs better than the conservative Q-function learning methods and is strongly competitive among recent SOTA methods.

## 1 Introduction

Reinforcement Learning (RL) learns to act by interacting with the environment and has shown great success in various tasks. However, in many real-world situations, it is impossible to learn from scratch online as exploration is often risky and unsafe. Instead, offline RL([1, 2]) avoids this problem by learning the policy solely from historical data. Yet, simply applying standard online RL techniques to static datasets can lead to overestimated values and incorrect policy decisions when faced with unfamiliar or out-of-distribution (OOD) scenarios.

---

[*]Work done during the internship at Microsoft. † Work done during full-time employment at Microsoft.

37th Conference on Neural Information Processing Systems (NeurIPS 2023).

Recently, the principle of conservative value estimation has been introduced to tackle challenges in offline RL[3, 4, 5]. Prior methods, e.g., CQL(Conservative Q-Learning [4]), avoid the value over-estimation problem by systematically underestimating the Q values of OOD actions on the states in the dataset. In practice, it is often too pessimistic and thus leads to overly conservative algorithms. COMBO [6] leverages a learned dynamic model to augment data in an interpolation way. This process helps derive a Q function that's less conservative than CQL, potentially leading to more optimal policies.

In this paper, we propose CSVE (Conservative State Value Estimation), a novel offline RL approach. Unlike the above methods that estimate conservative values by penalizing the Q-function for OOD actions, CSVE directly penalizes the V-function for OOD states. We theoretically demonstrate that CSVE provides tighter bounds on in-distribution state values in expectation than CQL, and same bounds as COMBO but under more general discounted state distributions, which potentially enhances policy optimization in the data support. Our main contributions include:

- The conservative state value estimation with related theoretical analysis. We prove that it lower bounds the real state values in expectation over any state distribution that is used to sample OOD states and is up-bounded by the real state values in expectation over the marginal state distribution of the dataset plus a constant term depending on sampling errors. Compared to prior work, it enhances policy optimization with conservative value guarantees.

- A practical actor-critic algorithm implemented CSVE. The critic undertakes conservative state value estimation, while the actor uses advantage-weighted regression(AWR) and explores states with conservative value guarantee to improve policy. In particular, we use a dynamics model to sample OOD states that are directly reachable from the dataset, for efficient value penalizing and policy exploring.

- Experimental evaluation on continuous control tasks of Gym [7] and Adroit [8] in D4RL [9] benchmarks, showing that CSVE performs better than prior methods based on conservative Q-value estimation, and is strongly competitive among main SOTA algorithms.

## 2   Preliminaries

**Offline Reinforcement Learning.** Consider the Markov Decision Process $M := (\mathcal{S}, \mathcal{A}, P, r, \rho, \gamma)$, which comprises the state space $\mathcal{S}$, the action space $\mathcal{A}$, the transition model $P : \mathcal{S} \times \mathcal{A} \to \Delta(\mathcal{S})$, the reward function $r : \mathcal{S} \times \mathcal{A} \to \mathbb{R}$, the initial state distribution $\rho$ and the discount factor $\gamma \in (0, 1]$. A stochastic policy $\pi : \mathcal{S} \to \mathcal{A}$ selects an action probabilistically based on the current state. A transition is the tuple $(s_t, a_t, r_t, s_{t+1})$ where $a_t \sim \pi(\cdot|s_t)$, $s_{t+1} \sim P(\cdot|s_t, a_t)$, and $r_t = r(s_t, a_t)$. It's assumed that the reward values adhere to $|r(s, a)| \leq R_{max}, \forall s, a$. A trajectory under $\pi$ is the random sequence $\tau = (s_0, a_0, r_0, s_1, a_1, r_1, \ldots, s_T)$ which consists of continuous transitions starting from $s_0 \sim \rho$.

Standard RL is to learn a policy $\pi \in \Pi$ that maximize the expected cumulative future rewards, represented as $J_\pi(M) = \mathbb{E}_{M,\pi}[\sum_{t=0}^\infty \gamma^t r_t]$, through active interaction with the environment $M$. At any time $t$, for the policy $\pi$, the value function of state is defined as $V^\pi(s) := \mathbb{E}_{M,\pi}[\sum_{k=0}^\infty \gamma^{t+k} r_{t+k}|s_t = s]$, and the Q value function is $Q^\pi(s, a) := \mathbb{E}_{M,\pi}[\sum_{k=0}^\infty \gamma^{t+k} r_{t+k}|s_t = s, a_t = a]$. The Bellman operator is a function projection: $\mathcal{B}^\pi Q(s, a) := r(s, a) + \gamma \mathbb{E}_{s' \sim P(\cdot|s,a), a' \sim \pi(\cdot|s')}[Q(s', a')]$, or $\mathcal{B}^\pi V(s) := \mathbb{E}_{a \sim \pi(\cdot|s)}[r(s, a) + \gamma \mathbb{E}_{s' \sim P(\cdot|s,a)}[V(s')]]$, resulting initerative value updates. Bellman consistency implies that $V^\pi(s) = \mathcal{B}^\pi V^\pi(s), \forall s$ and $Q^\pi(s) = \mathcal{B}^\pi Q^\pi(s, a), \forall s, a$. When employing function approximation in practice, the empirical Bellman operator $\hat{\mathcal{B}}^\pi$ is used, wherein the aforementioned expectations are estimated with data. Offline RL aims to learn the policy $\pi$ from a static dataset $D = \{(s, a, r, s')\}$ made up of transitions collected by any behavior policy, with the objective of performing well in the online setting. Note that, unlike standard online RL, offline RL does not interact with the environment during the learning process.

**Conservative Value Estimation.** One main challenge in offline RL arises from the over-estimation of values due to extrapolation in unseen states and actions. Such overestimation can lead to the deterioration of the learned policy. To address this issue, conservatism or pessimism is employed in value estimation. For instance, CQL learns a conservative Q-value function by penalizing the value

of unseen actions:

$$\hat{Q}^{k+1} \leftarrow \underset{Q}{\arg\min} \; \frac{1}{2}\mathbb{E}_{s,a,s'\sim D}[(Q(s,a)-\hat{\beta}_\pi \hat{Q}^k(s,a))^2]+\alpha\,(\mathbb{E}_{\substack{s\sim D \\ a\sim\mu(\cdot|s)}}[Q(s,a)]-\mathbb{E}_{\substack{s\sim D \\ a\sim\hat{\pi}_\beta(\cdot|s)}}[Q(s,a)])$$
(1)

where $\hat{\pi}_\beta$ and $\pi$ are the behaviour policy and learnt policy separately, $\mu$ is an arbitrary policy different from $\hat{\pi}_\beta$, and $\alpha$ represents the factor for balancing conservatism.

**Constrained Policy Optimization.** To address the issues of distribution shift between the learning policy and the behavior policy, one approach is to constrain the learning policy close to the behavior policy [10, 11, 12, 13, 1]. As an example, AdvantageWeighted Regression(AWR)[14, 12] employs an implicit KL divergence to regulate the distance between policies:

$$\pi^{k+1} \leftarrow \underset{\pi}{\arg\max} \; \mathbb{E}_{s,a\sim D}\left[\frac{\log\pi(a|s)}{Z(s)}\exp\left(\frac{1}{\lambda}A^{\pi^k}(s,a)\right)\right]$$

Here, $A^{\pi^k}$ is the advantage of policy $\pi^k$, and $Z$ serves as the normalization constant for $s$.

**Model-based Offline RL.** In RL, the model is an approximation of the MDP $M$. Such a model is denoted as $\hat{M} := (\mathcal{S}, \mathcal{A}, \hat{P}, \hat{r}, \rho, \gamma)$, with $\hat{P}$ and $\hat{r}$ being approximation of $P$ and $r$ respectively. Within offline RL, the model is commomly used to augment data [15, 6] or act as a surrogate of the real environment during interaction [16].However, such practices can inadvertently introduce bootstrapped errors over extended horizons[17]. In this paper, we restrict the use of the model to one-step sampling on the next states that are approximately reachable from the dataset.

## 3 Conservative State Value Estimation

In the offline setting, the value overestimation is a major problem resulting in failure of learning a reasonable policy [13, 1]. In contrast to prior works[4, 6] that get conservative value estimation via penalizing Q function for OOD state-action pairs, we directly penalize V function for OOD states. Our approach provides several novel theoretic results that allow better trade-off of conservative value estimation and policy improvement. All proofs of our theorems can be found in Appendix A.

### 3.1 Conservative Off-policy Evaluation

We aim to conservatively estimate the value of a target policy using a dataset to avoid overestimation of OOD states. To achieve this, we penalize V-values evaluated on states that are more likely to be OOD and increase the V-values on states that are in the distribution of the dataset. This adjustment is made iteratively::

$$\hat{V}^{k+1} \leftarrow \underset{V}{\arg\min}\; \frac{1}{2}\mathbb{E}_{s\sim d_u(s)}[(\hat{\mathcal{B}}^\pi\hat{V}^k(s) - V(s))^2] + \alpha(\mathbb{E}_{s'\sim d(s)}V(s') - \mathbb{E}_{s\sim d_u(s)}V(s)) \quad (2)$$

where $d_u(s)$ is the discounted state distribution of D, $d(s)$ is any state distribution, and $\hat{\mathcal{B}}^\pi$ is the empirical Bellman operator (see appendix for more details). Considering the setting without function approximation, by setting the derivative of Eq. 2 as zero, we can derive the V function using approximate dynamic programming at iteration $k$::

$$\hat{V}^{k+1}(s) = \hat{\mathcal{B}}^\pi\hat{V}^k(s) - \alpha[\frac{d_{(s)}}{d_u(s)} - 1], \quad \forall s, k. \quad (3)$$

Denote the function projection on $\hat{V}^k$ in Eq. 3 as $\mathcal{T}^\pi$. We have Lemma 3.1, which ensures that $\hat{V}^k$ converges to a unique fixed point.

**Lemma 3.1.** *For any $d$ with $\mathrm{supp}\,d \subseteq \mathrm{supp}\,d_u$, $\mathcal{T}^\pi$ is a $\gamma$-contraction in $L_\infty$ norm.*

**Theorem 3.2.** *For any $d$ with $\mathrm{supp}\,d \subseteq \mathrm{supp}\,d_u$ ($d \neq d_u$), with a sufficiently large $\alpha$ (i.e., $\alpha \geq \mathbb{E}_{s\sim d(s)}\mathbb{E}_{a\sim\pi(a|s)}\frac{C_{r,t,\delta}R_{max}}{(1-\gamma)\sqrt{|D(s,a)|}}/\mathbb{E}_{s\sim d(s)}[\frac{d(s)}{d_u(s)} - 1])$ ), the expected value of the estimation $\hat{V}^\pi(s)$ under $d(s)$ is the lower bound of the true value, that is: $\mathbb{E}_{s\sim d(s)}[\hat{V}^\pi(s)] \leq \mathbb{E}_{s\sim d(s)}[V^\pi(s)]$.*

$\hat{V}^\pi(s) = \lim_{k\to\infty} \hat{V}^k(s)$ is the converged value estimation with the dataset $D$, and $\frac{C_{r,t,\delta}R_{max}}{(1-\gamma)\sqrt{|D(s,a)|}}$ is related to sampling error that arises when using the empirical operator instead of the Bellman operator. If the counts of each state-action pair is greater than zero, $|D(s,a)|$ denotes a vector of size $|\mathcal{S}||\mathcal{A}|$ containing counts for each state-action pair. If the counts of this state action pair is zero, the corresponding $\frac{1}{\sqrt{|D(s,a)|}}$ is a large yet finite value. We assume that with probability $\geq 1-\delta$, the sampling error is less than $\frac{C_{r,t,\delta}R_{max}}{(1-\gamma)\sqrt{|D(s,a)|}}$, while $C_{r,t,\delta}$ is a constant (See appendix for more details.) Note that if the sampling error can be disregarded, $\alpha > 0$ can ensure the lower bound results.

**Theorem 3.3.** *The expected value of the estimation, $\hat{V}^\pi(s)$, under the state distribution of the original dataset is the lower bound of the true value plus the term of irreducible sampling error. Formally:*
$$\mathbb{E}_{s\sim d_u(s)}[\hat{V}^\pi(s)] \leq \mathbb{E}_{s\sim d_u(s)}[V^\pi(s)] + \mathbb{E}_{s\sim d_u(s)}(I-\gamma P^\pi)^{-1}\mathbb{E}_{a\sim\pi(a|s)}\frac{C_{r,t,\delta}R_{max}}{(1-\gamma)\sqrt{|D(s,a)|}}.$$

where $P^\pi$ refers to the transition matrix coupled with policy $\pi$ (see Appendix for details).

Now we show that, during iterations, the gap between the estimated V-function values of in-distribution states and OOD states is higher compared to the true V-functions.

**Theorem 3.4.** *For any iteration $k$, given a sufficiently large $\alpha$, our method amplifies the difference in expected V-values between the selected state distribution and the dataset state distribution. This can be represented as:* $\mathbb{E}_{s\sim d_u(s)}[\hat{V}^k(s)] - \mathbb{E}_{s\sim d(s)}[\hat{V}^k(s)] > \mathbb{E}_{s\sim d_u(s)}[V^k(s)] - \mathbb{E}_{s\sim d(s)}[V^k(s)].$

Our approach, which penalizes the V-function for OOD states, promotes a more conservative estimate of a target policy's value in offline reinforcement learning. Consequently, our policy extraction ensures actions align with the dataset's distribution.

To apply our approach effectively in offline RL algorithms, the preceding theorems serve as guiding principles. Here are four key insights for practical use of Eq. 2:

**Remark 1.** According to Eq. 2, if $d = d_u$, the penalty for OOD states diminishes. This means that the policy will likely avoid states with limited data support, preventing it from exploring unseen actions in such states. While AWAC [12] employs this configuration, our findings indicate that by selecting a $d$, our method surpasses AWAC's performance.

**Remark 2.** Theorem 3.3 suggests that under $d_u$, the marginal state distribution of data, the expectation estimated value of $V^\pi$ is either lower than its true value or exceed it, but within a certain limit. This understanding drives our adoption of the advantage-weighted policy update, as illustrated in Eq. 9.

**Remark 3.** As per Theorem 3.2, the expected estimated value of a policy under $d$, which represents the discounted state distribution of any policy, must be a lower bound of its true value. Grounded in this theorem, our policy enhancement strategy merges an advantage-weighted update with an additional exploration bonus, showcased in Eq. 10.

**Remark 4.** Theorem 3.4 states $\mathbb{E}_{s\sim d(s)}[V^k(s)] - \mathbb{E}_{s\sim d(s)}[\hat{V}^k(s)] > \mathbb{E}_{s\sim d_u(s)}[V^k(s)] - \mathbb{E}_{s\sim d_u(s)}[\hat{V}^k(s)]$. In simpler terms, the underestimation of value is more pronounced under $d$. With the proper choice of $d$, we can confidently formulate a newer and potentially superior policy using $\hat{V}^k$. Our algorithm chooses the distribution of model predictive next-states as $d$, i.e., $s' \sim d$ is implemented by $s \sim D, a \sim \pi(\cdot|s), s' \sim \hat{P}(\cdot|s,a)$, which effectively builds a soft 'river' with low values encircling the dataset.

**Comparison with prior work:** CQL (Eq.1), which penalizes Q-function of OOD actions, guarantees the lower bounds on state-wise value estimation: $\hat{V}^\pi(s) = E_{\pi(a|s)}(\hat{Q}^\pi(s,a)) \leq E_{\pi(a|s)}(Q^\pi(s,a)) = V^\pi(s)$ for all $s \in D$. COMBO, which penalizes the Q-function for OOD states and actions of interpolation of history data and model-based roll-outs, guarantees the lower bound of state value expectation: $\mathbb{E}_{s\sim\mu_0}[\hat{V}^\pi(s)] \leq \mathbb{E}_{s\sim\mu_0}[V^\pi(s)]$ where $\mu_0$ is the initial state distribution (Remark 1, section A.2 of COMBO [6]); which is a special case of our result in Theorem 3.2 when $d = \mu_0$. Both CSVE and COMBO intend to enhance performance by transitioning from individual state values to expected state values. However, CSVE offers the same lower bounds but under a more general state distribution. Note that $\mu_0$ depends on the environment or the dynamic model during offline training. CSVE's flexibility, represented by $d$, ensures conservative guarantees across any discounted state distribution of the learned policy, emphasizing a preference for penalizing $V$ over the Q-function.

## 3.2 Safe Policy Improvement Guarantees

Now we show that our method has the safe policy improvement guarantees against the data-implied behaviour policy. We first show that our method optimizes a penalized RL empirical objective:

**Theorem 3.5.** *Let $\hat{V}^\pi$ be the fixed point of Eq. 3, then $\pi^*(a|s) = \arg\max_\pi \hat{V}^\pi(s)$ is equivalently obtained by solving:*

$$\pi^* \leftarrow \arg\max_\pi J(\pi, \hat{M}) - \frac{\alpha}{1-\gamma} \mathbb{E}_{s \sim d_{\hat{M}}^\pi} \left[ \frac{d(s)}{d_u(s)} - 1 \right]. \tag{4}$$

Building upon Theorem 3.5, we show that our method provides a $\zeta$-safe policy improvement over $\pi_\beta$.

**Theorem 3.6.** *Let $\pi^*(a|s)$ be the policy obtained in Eq. 4. Then, it is a $\zeta$-safe policy improvement over $\hat{\pi}^\beta$ in the actual MDP M, i.e., $J(\pi^*, M) \geq J(\hat{\pi}^\beta, M) - \zeta$ with high probability 1- $\delta$, where $\zeta$ is given by:*

$$\zeta = 2 \left( \frac{C_{r,\delta}}{1-\gamma} + \frac{\gamma R_{max} C_{T,\delta}}{(1-\gamma)^2} \right) \mathbb{E}_{s \sim d_{\hat{M}}^\pi(s)} \left[ c \sqrt{\mathbb{E}_{a \sim \pi(a|s)} \left[ \frac{\pi(a|s)}{\pi_\beta(a|s)} \right]} \right]$$

$$- \underbrace{(J(\pi^*, \hat{M}) - J(\hat{\pi}_\beta, \hat{M}))}_{\geq \alpha \frac{1}{1-\gamma} \mathbb{E}_{s \sim d_{\hat{M}}^\pi(s)} [\frac{d(s)}{d_u(s)} - 1]} \text{ where } c = \sqrt{|\mathcal{A}|} / \sqrt{|\mathcal{D}(s)|}.$$

## 4 Methodology

In this section, we propose a practical actor-critic algorithm that employs CSVE for value estimation and extends Advantage Weighted Regression[18] with out-of-sample state exploration for policy improvement. In particular, we adopt a dynamics model to sample OOD states during conservative value estimation and exploration during policy improvement. The implementation details are in Appendix B. Besides, we discuss the general technical choices of applying CSVE into algorithms.

### 4.1 Conservative Value Estimation

Given a dataset $D$ acquired by the behavior policy $\pi_\beta$, our objective is to estimate the value function $V^\pi$ for a target policy $\pi$. As stated in section 3, to prevent the value overestimation, we learn a conservative value function $\hat{V}^\pi$ that lower bounds the real values of $\pi$ by adding a penalty for OOD states within the Bellman projection sequence. Our method involves iterative updates of Equations 5 - 7, where $\overline{\hat{Q}^k}$ is the target network of $\hat{Q}^k$.

$$\hat{V}^{k+1} \leftarrow \arg\min_V L_V^\pi(V; \overline{\hat{Q}^k}) \tag{5}$$

$$= \mathbb{E}_{s \sim D} \left[ (\mathbb{E}_{a \sim \pi(\cdot|s)}[\overline{\hat{Q}^k}(s,a)] - V(s))^2 \right] + \alpha \left( \mathbb{E}_{\substack{s \sim D, a \sim \pi(\cdot|s) \\ s' \sim \hat{P}(s,a)}}[V(s')] - \mathbb{E}_{s \sim D}[V(s)] \right)$$

$$\hat{Q}^{k+1} \leftarrow \arg\min_Q L_Q^\pi(Q; \hat{V}^{k+1}) = \mathbb{E}_{s,a,s' \sim D} \left[ \left( r(s,a) + \gamma \hat{V}^{k+1}(s') - Q(s,a) \right)^2 \right] \tag{6}$$

$$\overline{\hat{Q}^{k+1}} \leftarrow (1-\omega)\overline{\hat{Q}^k} + \omega \hat{Q}^{k+1} \tag{7}$$

The RHS of Eq. 5 is an approximation of Eq. 2, with the first term representing the standard TD error. In this term, the target state value is estimated by taking the expectation of $\overline{\hat{Q}^k}$ over $a \sim \pi$, and the second term penalizes the value of OOD states. In Eq. 6, the RHS is TD errors estimated on transitions in the dataset $D$. Note that the target term is the sum of the reward $r(s,a)$ and the next step state's value $\hat{V}^{k+1}(s')$. In Eq. 7, the target Q values are updated with a soft interpolation factor $\omega \in (0,1)$. $\overline{\hat{Q}^k}$ changes slower than $\hat{Q}^k$, which makes the TD error estimation in Eq. 5 more stable.

**Constrained Policy.** Note that in RHS of Eq. 5, we use $a \sim \pi(\cdot|s)$ in expectation. To safely estimate the target value of $V(s)$ by $\mathbb{E}_{a \sim \pi(\cdot|s)}[\overline{\hat{Q}}(s,a)]$, we almost always requires $\text{supp}(\pi(\cdot|s)) \subset$

$\text{supp}(\pi_\beta(\cdot|s))$. We achieve this by the *advantage weighted policy update*, which forces $\pi(\cdot|s)$ to have significant probability mass on actions taken by $\pi_\beta$ in data, as detailed in section 3.2.

**Model-based OOD State Sampling.** In Eq. 5, we implement the state sampling process $s' \sim d$ in Eq. 2 as a flow of $\{s \sim D; a \sim \pi(a|s), s' \sim \hat{P}(s'|s,a)\}$, that is the distribution of the predictive next-states from $D$ by following $\pi$. This approach proves beneficial in practice. On the one hand, this method is more efficient as it samples only the states that are approximately reachable from $D$ by one step, rather than sampling the entire state space. On the other hand, we only need the model to do one-step prediction such that it introduces no bootstrapped errors from long horizons. Following previous work [17, 15, 6], We use an ensemble of deep neural networks, represented as $p\theta^1, \ldots, p\theta^B$, to implement the probabilistic dynamics model. Each neural network produces a Gaussian distribution over the next state and reward: $P_\theta^i(s_{t+1}, r|s_t, a_t) = \mathcal{N}(u_\theta^i(s_t, a_t), \sigma_\theta^i(s_t, a_t))$.

**Adaptive Penalty Factor $\alpha$.** The pessimism level is controlled by the parameter $\alpha \geq 0$. In practice, we set $\alpha$ adaptive during training as follows, which is similar to that in CQL([4])

$$\max_{\alpha \geq 0} [\alpha(\mathbb{E}_{s' \sim d}[V_\psi(s')] - \mathbb{E}_{s \sim D}[V_\psi(s)] - \tau)], \tag{8}$$

where $\tau$ is a budget parameter. If the expected difference in V-values is less than $\tau$, $\alpha$ will decrease. Otherwise, $\alpha$ will increase, penalizing the OOD state values more aggressively.

### 4.2 Advantage Weighted Policy Update

After learning the conservative $\hat{V}^{k+1}$ and $\hat{Q}^{k+1}$ (or $\hat{V}^\pi$ and $\hat{Q}^\pi$ when the values have converged), we improve the policy by the following advantage weighted update [12].

$$\pi \leftarrow \arg\min_{\pi'} L_\pi(\pi') = - \mathop{\mathbb{E}}_{s,a \sim D} \left[ \log \pi'(a|s) \exp\left( \beta \hat{A}^{k+1}(s,a) \right) \right] \tag{9}$$

where $\hat{A}^{k+1}(s,a) = \hat{Q}^{k+1}(s,a) - \hat{V}^{k+1}(s)$. Eq.9 updates the policy $\pi$ by applying a weighted maximum likelihood method. This is computed by re-weighting state-action samples in $D$ using the estimated advantage $\hat{A}^{k+1}$. It avoids explicit estimation of the behavior policy, and its resulting sampling errors, which is an important issue in offline RL [12, 4].

**Implicit policy constraints.** We adopt the advantage-weighted policy update which imposes an implicit KL divergence constraint between $\pi$ and $\pi_\beta$. This policy constraint is necessary to guarantee that the next state $s'$ in Eq. 5 can be safely generated through policy $\pi$. As derived in [12] (Appendix A), Eq. 9 is a parametric solution of the following problem (where $\epsilon$ depends on $\beta$):

$$\max_{\pi'} \mathbb{E}_{a \sim \pi'(\cdot|s)}[\hat{A}^{k+1}(s,a)]$$

$$s.t. \ \mathrm{D_{KL}}(\pi'(\cdot|s) \ || \ \pi_\beta(\cdot|s)) \leq \epsilon, \quad \int_a \pi'(a|s)da = 1.$$

Note that $\mathrm{D_{KL}}(\pi' \ || \ \pi_\beta)$ is a reserved KL divergence with respect to $\pi'$, which is mode-seeking [19]. When treated as Lagrangian it forces $\pi'$ to allocate its probability mass to the maximum likelihood supports of $\pi_\beta$, re-weighted by the estimated advantage. In other words, for the space of $A$ where $\pi_\beta(\cdot|s)$ has no samples, $\pi'(\cdot|s)$ has almost zero probability mass too.

**Model-based Exploration on Near States.** As suggested by remarks in Section 3.1, in practice, allowing the policy to explore the predicted next states transition ($s \sim D$) following $a \sim \pi'(\cdot|s)$) leads to better test performance. With this kind of exploration, the policy is updated as follows:

$$\pi \leftarrow \arg\min_{\pi'} L_\pi(\pi') - \lambda \mathop{\mathbb{E}}_{\substack{s \sim D, a \sim \pi'(s) \\ s' \sim \hat{P}(s,a)}} \left[ r(s,a) + \gamma \hat{V}^{k+1}(s') \right]. \tag{10}$$

The second term is an approximation to $E_{s \sim d_\pi(s)}[V^\pi(s)]$. The optimization of this term involves calculating the gradient through the learned dynamics model. This is achieved by employing analytic gradients through the learned dynamics to maximize the value estimates. It is important to note that the value estimates rely on the reward and value predictions, which are dependent on the imagined states and actions. As all these steps are implemented using neural networks, the gradient is analytically computed using stochastic back-propagation, a concept inspired by Dreamer[20]. We adjust the value of $\lambda$, a hyper-parameter, to balance between optimistic policy optimization (in maximizing V) and the constrained policy update (as indicated by the first term).

Table 1: Performance comparison on Gym control tasks v2. The results of CSVE are over ten seeds and we reimplement AWAC using d3rlpy. Results of IQL, TD3-BC, and PBRL are from their original papers ( Table 1 in [21], Table C.3 in [22], and Table 1 in [10] respectively). Results of COMBO and CQL are from the reproduction results in [23] (Table 1) and [10] respectively, since their original results were reported on v0 datasets.

| | | AWAC | CQL | CQL-AWR | COMBO | IQL | TD3-BC | PBRL | CSVE |
|---|---|---|---|---|---|---|---|---|---|
| Random | HalfCheetah | 13.7 | $17.5 \pm 1.5$ | $16.9 \pm 1.5$ | 38.8 | 18.2 | $11.0 \pm 1.1$ | $13.1 \pm 1.2$ | $26.8 \pm 1.5$ |
| | Hopper | 8.7 | $7.9 \pm 0.4$ | $8.7 \pm 0.5$ | 17.9 | 16.3 | $8.5 \pm 0.6$ | $31.6 \pm 0.3$ | $26.1 \pm 7.6$ |
| | Walker2D | 2.2 | $5.1 \pm 1.3$ | $0.0 \pm 1.6$ | 7.0 | 5.5 | $1.6 \pm 1.7$ | $8.8 \pm 6.3$ | $6.2 \pm 0.8$ |
| Medium | HalfCheetah | 50.0 | $47.0 \pm 0.5$ | $50.9 \pm 0.6$ | 54.2 | 47.4 | $48.3 \pm 0.3$ | $58.2 \pm 1.5$ | $48.4 \pm 0.3$ |
| | Hopper | 97.5 | $53.0 \pm 28.5$ | $25.7 \pm 37.4$ | 94.9 | 66.3 | $59.3 \pm 4.2$ | $81.6 \pm 14.5$ | $96.7 \pm 5.7$ |
| | Walker2D | 89.1 | $73.3 \pm 17.7$ | $62.4 \pm 24.4$ | 75.5 | 78.3 | $83.7 \pm 2.1$ | $90.3 \pm 1.2$ | $83.2 \pm 1.0$ |
| Medium Replay | HalfCheetah | 44.9 | $45.5 \pm 0.7$ | $40.0 \pm 0.4$ | 55.1 | 44.2 | $44.6 \pm 0.5$ | $49.5 \pm 0.8$ | $54.5 \pm 0.6$ |
| | Hopper | 99.4 | $88.7 \pm 12.9$ | $91.0 \pm 13.0$ | 73.1 | 94.7 | $60.9 \pm 18.8$ | $100.7 \pm 0.4$ | $91.7 \pm 0.2$ |
| | Walker2D | 80.0 | $83.3 \pm 2.7$ | $66.7 \pm 12.1$ | 56.0 | 73.9 | $81.8 \pm 5.5$ | $86.2 \pm 3.4$ | $78.0 \pm 1.5$ |
| Medium Expert | HalfCheetah | 62.8 | $75.6 \pm 25.7$ | $73.4 \pm 2.0$ | 90.0 | 86.7 | $90.7 \pm 4.3$ | $93.1 \pm 0.2$ | $93.1 \pm 0.3$ |
| | Hopper | 87.2 | $105.6 \pm 12.9$ | $102.2 \pm 7.7$ | 111.1 | 91.5 | $98.0 \pm 9.4$ | $111.2 \pm 0.7$ | $94.1 \pm 3.0$ |
| | Walker2D | 109.8 | $107.9 \pm 1.6$ | $98.0 \pm 21.7$ | 96.1 | 109.6 | $110.1 \pm 0.5$ | $109.8 \pm 0.2$ | $109.0 \pm 0.1$ |
| Expert | HalfCheetah | 20.0 | $96.3 \pm 1.3$ | $87.3 \pm 8.1$ | - | 94.6 | $96.7 \pm 1.1$ | $96.2 \pm 2.3$ | $93.8 \pm 0.1$ |
| | Hopper | 111.6 | $96.5 \pm 28.0$ | $110.0 \pm 2.5$ | - | 109.0 | $107.8 \pm 7$ | $110.4 \pm 0.3$ | $111.3 \pm 0.6$ |
| | Walker2D | 110.6 | $108.5 \pm 0.5$ | $75.1 \pm 60.7$ | - | 109.4 | $110.2 \pm 0.3$ | $108.8 \pm 0.2$ | $108.5 \pm 0.1$ |
| Average | | 65.8 | 67.4 | 60.6 | 64.1 | 69.7 | 67.5 | 76.7 | 74.8 |

## 4.3 Discussion on implementation choices

Now we examine the technical considerations for implementing CSVE in a practical algorithm.

**Constraints on Policy Extraction.** It is important to note that the state value function alone does not suffice to directly derive a policy. There are two methods for extracting a policy from CSVE. The first method is model-based planning, i.e., $\pi \leftarrow \arg\max_{\pi} \mathbb{E}_{s \sim d, a \sim \pi(\cdot|s)}[\hat{r}(s, a) + \gamma \, \mathbb{E}_{s' \sim \hat{P}(s,a)}[V(s')]$, which involves finding the policy that maximizes the expected future return. However, this method heavily depends on the accuracy of a model and is difficult to implement in practice. As an alternative, we suggest the second method, which learns a Q value or advantage function from the V value function and experience data, and then extracts the policy. Note that CSVE provides no guarantees for conservative estimation on OOD actions, which can cause normal policy extraction methods such as SAC to fail. To address this issue, we adopt policy constraint techniques. On the one hand, during the value estimation in Eq.5, all current states are sampled from the dataset, while the policy is constrained to be close to the behavior policy (ensured via Eq.9). On the other hand, during the policy learning in Eq.10, we use AWR [18] as the primary policy extraction method (first term of Eq.10), which implicitly imposes policy constraints and the additional action exploration (second term of Eq.10) is strictly applied to states in the dataset. This exploration provides a bonus to actions that: (1) themselves and their model-predictive next-states are both close to the dataset (ensured by the dynamics model), and (2) their values are favorable even with conservatism.

**Taking Advantage of CSVE.** As outlined in Section 3.1, CSVE allows for a more relaxed lower bound on conservative value estimation compared to conservative Q values, providing greater potential for improving the policy. To take advantage of this, the algorithm should enable exploration of out-of-sample but in-distribution states, as described in Section 3. In this paper, we use a deep ensemble dynamics model to support this speculative state exploration, as shown in Eq. 10. The reasoning behind this is as follows: for an in-data state $s$ and any action $a \sim \pi(\cdot|s)$, if the next state $s'$ is in-data or close to the data support, its value is reasonably estimated, and if not, its value has been penalized according to Eq.5. Additionally, the deep ensemble dynamics model captures epistemic uncertainty well, which can effectively cancel out the impact of rare samples of $s'$. By utilizing CSVE, our algorithm can employ the speculative interpolation to further improve the policy. In contrast, CQL and AWAC do not have this capability for such enhanced policy optimization.

Table 2: Performance comparison on Adroit tasks. The results of CSVE are over ten seeds. Results of IQL are from Table 3 in [21] and results of other algorithms are from Table 4 in [10].

|  |  | AWAC | BC | BEAR | UWAC | CQL | CQL-AWR | IQL | PBRL | CSVE |
|---|---|---|---|---|---|---|---|---|---|---|
| Human | Pen | 18.7 | 34.4 | -1.0 | $10.1 \pm 3.2$ | 37.5 | $8.4 \pm 7.1$ | 71.5 | $35.4 \pm 3.3$ | $106.2 \pm 5.0$ |
|  | Hammer | -1.8 | 1.5 | 0.3 | $1.2 \pm 0.7$ | 4.4 | $0.3 \pm 0.0$ | 1.4 | $0.4 \pm 0.3$ | $3.5 \pm 2.6$ |
|  | Door | -1.8 | 0.5 | $-0.3$ | $0.4 \pm 0.2$ | 9.9 | $3.5 \pm 1.8$ | 4.3 | $0.1 \pm 0.0$ | $2.8 \pm 2.4$ |
|  | Relocate | -0.1 | 0.0 | -0.3 | $0.0 \pm 0.0$ | 0.2 | $0.1 \pm 0.0$ | 0.1 | $0.0 \pm 0.0$ | $0.1 \pm 0.0$ |
| Cloned | Pen | 27.2 | 56.9 | 26.5 | $23.0 \pm 6.9$ | 39.2 | $29.3 \pm 7.1$ | 37.3 | $74.9 \pm 9.8$ | $54.5 \pm 5.4$ |
|  | Hammer | -1.8 | 0.8 | 0.3 | $0.4 \pm 0.0$ | 2.1 | $0.31 \pm 0.06$ | 2.1 | $0.8 \pm 0.5$ | $0.5 \pm 0.2$ |
|  | Door | -2.1 | -0.1 | $-0.1$ | $0.0 \pm 0.0$ | 0.4 | $-0.2 \pm 0.1$ | 1.6 | $4.6 \pm 4.8$ | $1.2 \pm 1.0$ |
|  | Relocate | -0.4 | -0.1 | -0.3 | $-0.3 \pm 0.2$ | -0.1 | $-0.3 \pm 0.0$ | 0.0 | $-0.1 \pm 0.0$ | $-0.3 \pm 0.0$ |
| Expert | Pen | 60.9 | 85.1 | 105.9 | $98.2 \pm 9.1$ | 107.0 | $47.1 \pm 6.8$ | 117.2 | $135.7 \pm 3.4$ | $144.0 \pm 9.4$ |
|  | Hammer | 31.0 | 125.6 | 127.3 | $107.7 \pm 21.7$ | 86.7 | $0.2 \pm 0.0$ | 124.1 | $127.5 \pm 0.2$ | $126.5 \pm 0.3$ |
|  | Door | 98.1 | 34.9 | 103.4 | $104.7 \pm 0.4$ | 101.5 | $85.0 \pm 15.9$ | 105.2 | $95.7 \pm 12.2$ | $104.2 \pm 0.8$ |
|  | Relocate | 49.0 | 101.3 | 98.6 | $105.5 \pm 3.2$ | 95.0 | $7.2 \pm 12.5$ | 105.9 | $84.5 \pm 12.2$ | $102.9 \pm 0.9$ |
| Average |  | 23.1 | 36.7 | 38.4 | 37.6 | 40.3 | 15.1 | 47.6 | 46.6 | 53.8 |

## 5 Experiments

This section evaluates the effectiveness of our proposed CSVE algorithm for conservative value estimation in offline RL. In addition, we aim to compare the performance of CSVE with state-of-the-art (SOTA) algorithms. To achieve this, we conduct experimental evaluations on a variety of classic continuous control tasks of Gym[7] and Adroit[8] in the D4RL[9] benchmark.

Our compared baselines include: (1) CQL[4] and its variants, CQL-AWR (Appendix D.2) which uses AWR with extra in-sample exploration as policy extractor, COMBO[6] which extends CQL with model-based rollouts; (2) AWR variants, including AWAC[12] which is a special case of our algorithm with no value penalization (i.e., $d = d_u$ in Eq. 2) and exploration on OOD states, IQL[21] which adopts expectile-based conservative value estimation; (3) PBRL[10], a strong algorithm in offline RL, but is quite costly on computation since it uses the ensemble of hundreds of sub-models; (4) other SOTA algorithms with public performance results or high-quality open source implementations, including TD3-BC[22], UWAC[24] and BEAR[25]). Comparing with CQL variants allows us to investigate the advantages of conservative estimation on state values over Q values. By comparing with AWR variants, we distinguish the performance contribution of CSVE from the AWR policy extraction used in our implementation.

### 5.1 Overall Performance

**Evaluation on the Gym Control Tasks.** Our method, CSVE, was trained for 1 million steps and evaluated. The results are shown in Table 1.Compared to CQL, CSVE outperforms it in 11 out of 15 tasks, with similar performance on the remaining tasks. Additionally, CSVE shows a consistent advantage on datasets that were generated by following random or sub-optimal policies (random and medium). The CQL-AWR method showed slight improvement in some cases, but still underperforms compared to CSVE. When compared to COMBO, CSVE performs better in 7 out of 12 tasks and similarly or slightly worse on the remaining tasks, which highlights the effectiveness of our method's better bounds on V. Our method has a clear adcantage in extracting the best policy on medium and medium-expert tasks. Overall, our results provide empirical evidence that using conservative value estimation on states, rather than Q, leads to improved performance in offline RL. CSVE outperforms AWAC in 9 out of 15 tasks, demonstrating the effectiveness of our approach in exploring beyond the behavior policy. Additionally, our method excels in extracting the optimal policy on data with mixed policies (medium-expert) where AWAC falls short. In comparison to IQL, our method achieves higher scores in 7 out of 9 tasks and maintains comparable performance in the remaining tasks. Furthermore, despite having a significantly lower model capacity and computation cost, CSVE outperforms TD3-BC and is on par with PBRL. These results highlight the effectiveness of our conservative value estimation approach.

**Evaluation on the Adroit Tasks.** In Table 2, we report the final evaluation results after training 0.1 million steps. As shown, our method outperforms IQL in 8 out of 12 tasks, and is competitive with other algorithms on expert datasets. Additionally, we note that CSVE is the only method that

can learn an effective policy on the human dataset for the Pen task, while maintaining medium performance on the cloned dataset. Overall, our results empirically support the effectiveness of our proposed tighter conservative value estimation in improving offline RL performance.

## 5.2 Ablation Study

**Effect of Exploration on Near States.** We analyze the impact of varying the factor $\lambda$ in Eq. 10, which controls the intensity on such exploration. We investigated $\lambda$ values of $\{0.0, 0.1, 0.5, 1.0\}$ in the medium tasks, fixing $\beta = 0.1$. The results are plotted in Fig. 1. As shown in the upper figures, $\lambda$ has an obvious effect on policy performance and variances during training. With increasing $\lambda$ from 0, the converged performance gets better in general. However, when the value of $\lambda$ becomes too large (e.g., $\lambda = 3$ for hopper and walker2d), the performance may degrade or even collapse. We further investigated the $L_\pi$ loss as depicted in the bottom figures of Eq. 9, finding that larger $\lambda$ values negatively impact $L_\pi$; however, once $L_\pi$ converges to a reasonable low value, larger $\lambda$ values lead to performance improvement.

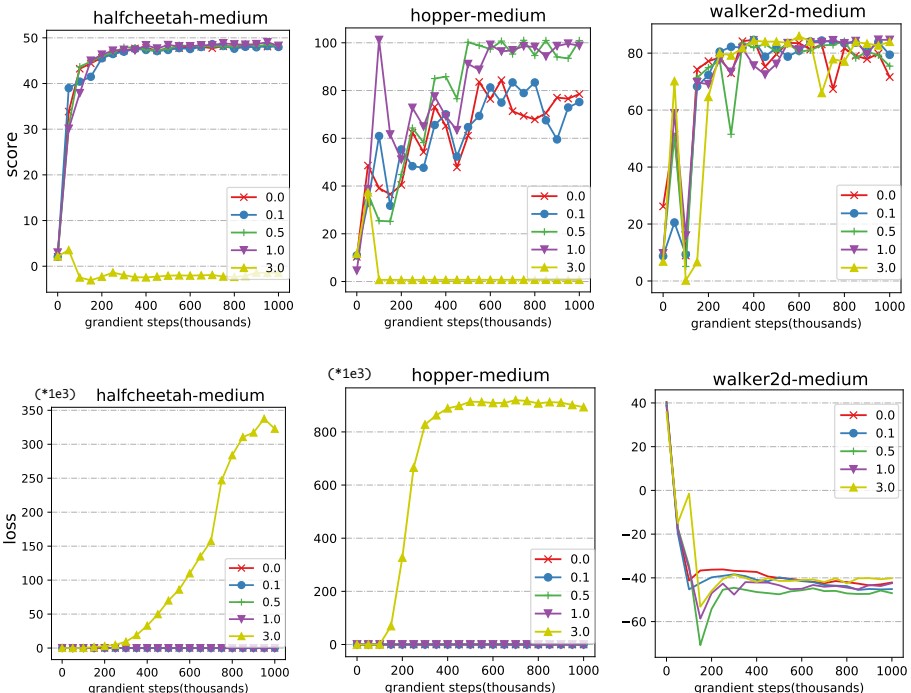

Figure 1: Effect of $\lambda$ to performance scores (upper figures) and $L_\pi$ losses (bottom figures) in Eq. 9 on medium tasks.

**Effect of In-sample Policy Optimization.** We examined the impact of varying the factor $\beta$ in Eq. 9 on the balance between behavior cloning and in-sample policy optimization. We tested different $\beta$ values on mujoco medium datasets, as shown in Fig.2. The results indicate that $\beta$ has a significant effect on the policy performance during training. Based on our findings, a value of $\beta = 3.0$ was found to be suitable for medium datasets. Additionally, in our implementation, we use $\beta = 3.0$ for random and medium tasks, and $\beta = 0.1$ for medium-replay, medium-expert, and expert datasets. More details can be found in the ablation study in the appendix.

## 6 Related work

The main idea behind offline RL algorithms is to incorporate conservatism or regularization into the online RL algorithms. Here, we briefly review prior work and compare it to our approach.

**Conservative Value Estimation:** Prior offline RL algorithms regularize the learning policy to be close to the data or to an explicitly estimated behavior policy. and penalize the exploration

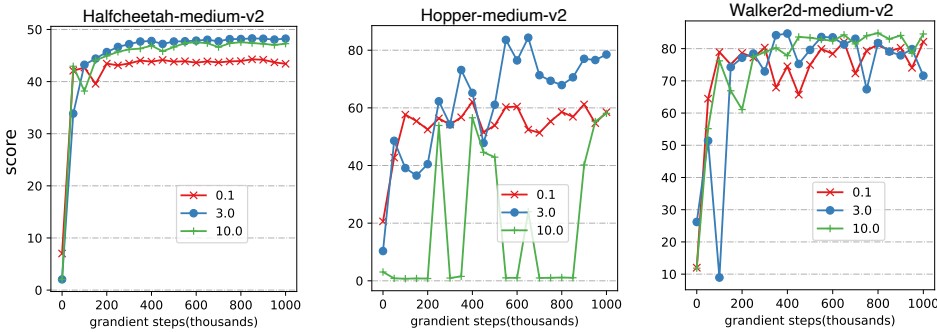

Figure 2: Effect of $\beta$ to performance scores on medium tasks.

ofthe OOD region, via distribution correction estimation [26, 27], policy constraints with support matching [11] and distributional matching [1, 25], applying policy divergence based penalty on Q-functions [28, 29] or uncertainty-based penalty [30] on Q-functions and conservative Q-function estimation [4]. Besides, model-based algorithms [15] directly estimate dynamics uncertainty and translate it into reward penalty. Different from this prior work that imposes conservatism on state-action pairs or actions, ours directly does such conservative estimation on states and requires no explicit uncertainty quantification.

**In-Sample Algorithms:** AWR [18] updates policy constrained on strictly in-sample states and actions, to avoid extrapolation on out-of-support points. IQL[21] uses expectile-based regression to do value estimation and AWR for its policy updates. AWAC[12], whose actor is AWR, is an actor-critic algorithm to accelerate online RL with offline data. The major drawback of AWR method when used for offline RL is that the in-sample policy learning limits the final performance.

**Model-Based Algorithms:** Model-based offline RL learns the dynamics model from the static dataset and uses it to quantify uncertainty [15], data augmentation [6] with roll-outs, or planning [16, 31]. Such methods typically rely on wide data coverage when planning and data augmentation with roll-outs, and low model estimation error when estimating uncertainty, which is difficult to satisfy in reality and leads to policy instability. Instead, we use the model to sample the next-step states only reachable from data, which has no such strict requirements on data coverage or model bias.

**Theoretical Results:** Our theoretical results are derived from conservative Q-value estimation (CQL) and safe policy improvement [32]. Compared to offline policy evaluation[33], which aims to provide a better estimation of the value function, we focus on providing a better lower bound. Additionally, hen the dataset is augmented with model-based roll-outs, COMBO [6] provides a more conservative yet tighter value estimation than CQL. CSVE achives the same lower bounds as COMBO but under more general state distributions.

# 7 Conclusions

In this paper, we propose CSVE, a new approach for offline RL based on conservative value estimation on states. We demonstrated how its theoretical results can lead to more effective algorithms. In particular, we develop a practical actor-critic algorithm, in which the critic achieves conservative state value estimation by incorporating the penalty of the model predictive next-states into Bellman iterations, and the actor does the advantage-weighted policy updates enhanced via model-based state exploration. Experimental evaluation shows that our method performs better than alternative methods based on conservative Q-function estimation and is competitive among the SOTA methods, thereby validating our theoretical analysis. Moving forward, we aim to delve deeper into designing more powerful algorithms grounded in conservative state value estimation.

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

# A  Proofs

We first redefine notation for clarity and then provide the proofs of the results from the main paper.

**Notation**. Let $k \in N$ denotes an iteration of policy evaluation(in Section 3.2). $V^k$ denotes the true, tabular (or functional) V-function iterate in the MDP. $\hat{V}^k$ denotes the approximate tabular (or functional) V-function iterate.

The empirical Bellman operator is defined as:

$$(\hat{\mathcal{B}}^\pi \hat{V}^k)(s) = E_{a \sim \pi(a|s)} \hat{r}(s,a) + \gamma \sum_{s'} E_{a \sim \pi(a|s)} \hat{P}(s'|s,a)[\hat{V}^k(s')] \tag{11}$$

where $\hat{r}(s,a)$ is the empirical average reward derived from the dataset when performing action $a$ at state $s$. The true Bellman operator is given by:

$$(\mathcal{B}^\pi V^k)(s) = E_{a \sim \pi(a|s)} r(s,a) + \gamma \sum_{s'} E_{a \sim \pi(a|s)} P(s'|s,a)[V^k(s')] \tag{12}$$

Now we first prove that the iteration in Eq.2 has a fixed point. Assume the state value function is lower bounded, i.e., $V(s) \geq C \ \forall s \in S$, then Eq.2 can always be solved with Eq.3. Thus, we only need to investigate the iteration in Eq.3.

Defining this iteration as a function operator $\mathcal{T}^\pi$ on $V$ and supposing that $\operatorname{supp} d \subseteq \operatorname{supp} d_u$, it's evident that the operator $\mathcal{T}^\pi$ displays a $\gamma$-contraction within $L_\infty$ norm where $\gamma$ is the discounting factor.

**Proof of Lemma 3.1:** Let $V$ and $V'$ be any two state value functions with the same support, i.e., $\operatorname{supp} V = \operatorname{supp} V'$.

$$
\begin{aligned}
|(\mathcal{T}^\pi V - \mathcal{T}^\pi V')(s)| &= \left| (\hat{\mathcal{B}}^\pi V(s) - \alpha[\frac{d_{(s)}}{d_u(s)} - 1]) - (\hat{\mathcal{B}}^\pi V'(s) - \alpha[\frac{d_{(s)}}{d_u(s)} - 1]) \right| \\
&= \left| \hat{\mathcal{B}}^\pi V(s) - \hat{\mathcal{B}}^\pi V'(s) \right| \\
&= |(E_{a \sim \pi(a|s)} \hat{r}(s,a) + \gamma E_{a \sim \pi(a|s)} \sum_{s'} \hat{P}(s'|s,a) V(s')) \\
&\quad - (E_{a \sim \pi(a|s)} \hat{r}(s,a) + \gamma E_{a \sim \pi(a|s)} \sum_{s'} \hat{P}(s'|s,a) V'(s'))| \\
&= \gamma \left| E_{a \sim \pi(a|s)} \sum_{s'} \hat{P}(s'|s,a)[V(s') - V'(s')] \right|
\end{aligned}
$$

$$
\begin{aligned}
||\mathcal{T}^\pi V - \mathcal{T}^\pi V'||_\infty &= \max_s |(\mathcal{T}^\pi V - \mathcal{T}^\pi V')(s)| \\
&= \max_s \gamma \left| E_{a \sim \pi(a|s)} \sum_{s'} \hat{P}(s'|s,a)[V(s') - V'(s')] \right| \\
&\leq \gamma E_{a \sim \pi(a|s)} \sum_{s'} \hat{P}(s'|s,a) \max_{s''} |V(s'') - V'(s'')| \\
&= \gamma \max_{s''} |V(s'') - V'(s'')| \\
&= \gamma ||(V - V')||_\infty
\end{aligned}
$$

$\square$

We provide a bound on the difference between the empirical Bellman operator and the true Bellman operator. Following previous work [4], we make the following assumptions. Let $P^\pi$ be the transition matrix associated with the policy, specifically, $P^\pi V(s) = E_{a' \sim \pi(a'|s'), s' \sim P(s'|s,a')}[V(s')]$

**Assumption A.1.** $\forall s, a \in \mathcal{M}$, the relationship below hold with at least a $(1-\delta)$ $(\delta \in (0,1))$ probability,

$$|r - r(s,a)| \leq \frac{C_{r,\delta}}{\sqrt{|D(s,a)|}}, ||\hat{P}(s'|s,a) - P(s'|s,a)||_1 \leq \frac{C_{P,\delta}}{\sqrt{|D(s,a)|}} \tag{13}$$

Given this assumption, the absolute difference between the empirical Bellman operator and the true one can be deduced as follows:

$$|(\hat{\mathcal{B}}^\pi)\hat{V}^k - (\mathcal{B}^\pi)\hat{V}^k)| = E_{a\sim\pi(a|s)}|r - r(s,a) + \gamma\sum_{s'} E_{a'\sim\pi(a'|s')}(\hat{P}(s'|s,a) - P(s'|s,a))[\hat{V}^k(s')]| \tag{14}$$

$$\leq E_{a\sim\pi(a|s)}|r - r(s,a)| + \gamma|\sum_{s'} E_{a'\sim\pi(a'|s')}(\hat{P}(s'|s,a') - P(s'|s,a'))[\hat{V}^k(s')]| \tag{15}$$

$$\leq E_{a\sim\pi(a|s)}\frac{C_{r,\delta} + \gamma C_{P,\delta}2R_{max}/(1-\gamma)}{\sqrt{|D(s,a)|}} \tag{16}$$

The error in estimation due to sampling can therefore be bounded by a constant, dependent on $C_{r,\delta}$ and $C_{t,\delta}$. We define this constant as $C_{r,T,\delta}$.

Thus we obtain:

$$\forall V, s \in D, |\hat{\mathcal{B}}^\pi V(s) - \mathcal{B}^\pi V(s)| \leq E_{a\sim\pi(a|s)}\frac{C_{r,t,\delta}}{(1-\gamma)\sqrt{|D(s,a)|}} \tag{17}$$

Next we provide an important lemma.

**Lemma A.2.** *(Interpolation Lemma) For any $f \in [0,1]$, and any given distribution $\rho(s)$, let $d_f$ be an f-interpolation of $\rho$ and $D$, i.e.,$d_f(s) := fd(s) + (1-f)\rho(s)$, let $v(\rho,f) := E_{s\sim\rho(s)}[\frac{\rho(s)-d(s)}{d_f(s)}]$, then $v(\rho,f)$ satisfies $v(\rho,f) \geq 0$.*

The proof can be found in [6]. By setting $f$ as 1, we have $E_{s\sim\rho(s)}[\frac{\rho(s)-d(s)}{d(s)}] > 0$.

**Proof of Theorem 3.2:** The V function of approximate dynamic programming in iteration $k$ can be obtained as:

$$\hat{V}^{k+1}(s) = \hat{\mathcal{B}}^\pi\hat{V}^k(s) - \alpha[\frac{d(s)}{d_u(s)} - 1] \forall s, k \tag{18}$$

The fixed point:

$$\hat{V}^\pi(s) = \hat{\mathcal{B}}^\pi\hat{V}^\pi(s) - \alpha[\frac{d(s)}{d_u(s)} - 1] \leq \mathcal{B}^\pi\hat{V}^\pi(s) + E_{a\sim\pi(a|s)}\frac{C_{r,t,\delta}R_{max}}{(1-\gamma)\sqrt{|D(s,a)|}} - \alpha[\frac{d(s)}{d_u(s)} - 1] \tag{19}$$

Thus we obtain:

$$\hat{V}^\pi(s) \leq V^\pi(s) + (I - \gamma P^\pi)^{-1}E_{a\sim\pi(a|s)}\frac{C_{r,t,\delta}R_{max}}{(1-\gamma)\sqrt{|D(s,a)|}} - \alpha(I - \gamma P^\pi)^{-1}[\frac{d(s)}{d_u(s)} - 1] \tag{20}$$

, where $P^\pi$ is the transition matrix coupled with the policy $\pi$ and $P^\pi V(s) = E_{a'\sim\pi(a'|s')s'\sim P(s'|s,a')}[V(s')]$.

Then the expectation of $V^\pi(s)$ under distribution $d(s)$ satisfies:

$$E_{s\sim d(s)}\hat{V}^\pi(s) \leq E_{s\sim d(s)}(V^\pi(s)) + E_{s\sim d(s)}(I - \gamma P^\pi)^{-1}E_{a\sim\pi(a|s)}\frac{C_{r,t,\delta}R_{max}}{(1-\gamma)\sqrt{|D(s,a)|}}$$

$$- \alpha\underbrace{E_{s\sim d(s)}(I - \gamma P^\pi)^{-1}[\frac{d(s)}{d_u(s)} - 1])}_{>0} \tag{21}$$

When $\alpha \geq \dfrac{E_{s\sim d(s)}E_{a\sim \pi(a|s)}\frac{C_{r,t,\delta}R_{max}}{(1-\gamma)\sqrt{|D(s,a)|}}}{E_{s\sim d(s)}[\frac{d(s)}{d_u(s)}-1])}$, $E_{s\sim d(s)}\hat{V}^\pi(s) \leq E_{s\sim d(s)}(V^\pi(s))$. $\qquad\square$

**Proof of Theorem 3.3:** The expectation of $V^\pi(s)$ under distribution $d(s)$ satisfies:

$$E_{s\sim d_u(s)}\hat{V}^\pi(s) \leq E_{s\sim d_u(s)}(V^\pi(s)) + E_{s\sim d_u(s)}(I-\gamma P^\pi)^{-1}E_{a\sim \pi(a|s)}\frac{C_{r,t,\delta}R_{max}}{(1-\gamma)\sqrt{|D(s,a)|}}$$
$$- \alpha E_{s\sim d_u(s)}(I-\gamma P^\pi)^{-1}[\frac{d(s)}{d_u(s)}-1]) \tag{22}$$

Noticed that the last term:

$$\sum_{s\sim d_u(s)}(\frac{d_f(s)}{d_u(s)}-1) = \sum_s d_u(s)(\frac{d_f(s)}{d_u(s)}-1) = \sum_s d_f(s) - \sum_s d_u(s) = 0 \tag{23}$$

We obtain that:

$$E_{s\sim d_u(s)}\hat{V}^\pi(s) \leq E_{s\sim d_u(s)}(V^\pi(s)) + E_{s\sim d_u(s)}(I-\gamma P^\pi)^{-1}E_{a\sim \pi(a|s)}\frac{C_{r,t,\delta}R_{max}}{(1-\gamma)\sqrt{|D(s,a)|}} \tag{24}$$

$\qquad\square$

**Proof of Theorem 3.4:** Recall that the expression of the V-function iterate is given by:

$$\hat{V}^{k+1}(s) = \mathcal{B}^{\pi^k}\hat{V}^k(s) - \alpha[\frac{d(s)}{d_u(s)}-1]\forall s, k \tag{25}$$

Now the expectation of $V^\pi(s)$ under distribution $d_u(s)$ is given by:

$$E_{s\sim d_u(s)}\hat{V}^{k+1}(s) = E_{s\sim d_u(s)}\left[\mathcal{B}^{\pi^k}\hat{V}^k(s) - \alpha[\frac{d(s)}{d_u(s)}-1]\right] = E_{s\sim d_u(s)}\mathcal{B}^{\pi^k}\hat{V}^k(s) \tag{26}$$

The expectation of $V^\pi(s)$ under distribution $d(s)$ is given by:

$$E_{s\sim d(s)}\hat{V}^{k+1}(s) = E_{s\sim d(s)}\mathcal{B}^{\pi^k}\hat{V}^k(s) - \alpha[\frac{d(s)}{d_u(s)}-1] = E_{s\sim d(s)}\mathcal{B}^{\pi^k}\hat{V}^k(s) - \alpha E_{s\sim d(s)}[\frac{d(s)}{d_u(s)}-1] \tag{27}$$

Thus we can show that:

$$E_{s\sim d_u(s)}\hat{V}^{k+1}(s) - E_{s\sim d(s)}\hat{V}^{k+1}(s) = E_{s\sim d_u(s)}\mathcal{B}^{\pi^k}\hat{V}^k(s) - E_{s\sim d(s)}\mathcal{B}^{\pi^k}\hat{V}^k(s) + \alpha E_{s\sim d(s)}[\frac{d(s)}{d_u(s)}-1]$$
$$= E_{s\sim d_u(s)}V^{k+1}(s) - E_{s\sim d(s)}V^{k+1}(s) - E_{s\sim d(s)}[\mathcal{B}^{\pi^k}(\hat{V}^k-V^k)]$$
$$+ E_{s\sim d_u(s)}[\mathcal{B}^{\pi^k}(\hat{V}^k-V^k)] + \alpha E_{s\sim d(s)}[\frac{d(s)}{d_u(s)}-1] \tag{28}$$

By choosing $\alpha$:

$$\alpha > \frac{E_{s\sim d(s)}[\mathcal{B}^{\pi^k}(\hat{V}^k-V^k)] - E_{s\sim d_u(s)}[\mathcal{B}^{\pi^k}(\hat{V}^k-V^k)]}{E_{s\sim d(s)}[\frac{d(s)}{d_u(s)}-1]} \tag{29}$$

We have $E_{s\sim d_u(s)}\hat{V}^{k+1}(s) - E_{s\sim d(s)}\hat{V}^{k+1}(s) > E_{s\sim d_u(s)}V^{k+1}(s) - E_{s\sim d(s)}V^{k+1}(s)$ hold. $\quad\square$

**Proof of Theorem 3.5:** $\hat{V}$ is obtained by solving the recursive Bellman fixed point equation in the empirical MDP, with an altered reward, $r(s,a) - \alpha[\frac{d(s)}{d_u(s)}-1]$, hence the optimal policy $\pi^*(a|s)$ obtained by optimizing the value under Eq. 3.5. $\qquad\square$

**Proof of Theorem 3.6:** The proof of this statement is divided into two parts. We first relates the return of $\pi^*$ in the empirical MDP $\hat{M}$ with the return of $\pi_\beta$, we can get:

$$J(\pi^*, \hat{M}) - \alpha\frac{1}{1-\gamma}\mathbb{E}_{s\sim d_{\hat{M}}^{\pi^*}(s)}[\frac{d(s)}{d_u(s)}-1] \geq J(\pi_\beta, \hat{M}) - 0 = J(\pi_\beta, \hat{M}) \tag{30}$$

---

**Algorithm 1** CSVE based Offline RL Algorithm

---

**Input:** data $D = \{(s, a, r, s')\}$
**Parametered Models:** $Q_\theta$, $V_\psi$, $\pi_\phi$, $Q_{\overline{\theta}}$, $M_\nu$
**Hyperparameters:** $\alpha, \lambda$, learning rates $\eta_\theta, \eta_\psi, \eta_\phi, \omega$
▷ *Train the transition model with the static dataset D*
$M_\nu \leftarrow train(D)$.
▷ *Train the conservative value estimation and policy functions*
Initialize function parameters $\theta_0, \psi_0, \phi_0, \overline{\theta}_0 = \theta_0$
**for** step $k = 1$ **to** $N$ **do**

$\quad \psi_k \leftarrow \psi_{k-1} - \eta_\psi \nabla_\psi L_V^\pi(V_\psi; \overline{\hat{Q}_{\theta_k}})$
$\quad \theta_k \leftarrow \theta_{k-1} - \eta_\theta \nabla_\theta L_Q^\pi(Q_\theta; \hat{V}_{\psi_k})$
$\quad \phi_k \leftarrow \phi_{k-1} - \eta_\phi \nabla_\phi L_\pi^+(\pi_\phi)$
$\quad \overline{\theta}_k \leftarrow \omega \overline{\theta}_{k-1} + (1 - \omega)\theta_k$

**end for**

---

The next step is to bound the difference between $J(\pi_\beta, \hat{M})$ and $J(\pi_\beta, M)$ and the difference between $J(\pi^*, \hat{M})$ and $J(\pi^*, M)$. We quote a useful lemma from [4] (Lemma D.4.1):

**Lemma A.3.** *For any MDP $M$, an empirical MDP $\hat{M}$ generated by sampling actions according to the behavior policy $\pi_\beta$ and a given policy $\pi$,*

$$|J(\pi, \hat{M}) - J(\pi, M)| \leq \left(\frac{C_{r,\delta}}{1-\gamma} + \frac{\gamma R_{max} C_{T,\delta}}{(1-\gamma)^2}\right) \mathbb{E}_{s \sim d_{\hat{M}}^{\pi^*}(s)}\left[\frac{\sqrt{|\mathcal{A}|}}{\sqrt{|\mathcal{D}(s)|}}\sqrt{E_{a \sim \pi(a|s)}\left(\frac{\pi(a|s)}{\pi_\beta(a|s)}\right)}\right] \quad (31)$$

Setting $\pi$ in the above lemma as $\pi_\beta$, we get:

$$|J(\pi_\beta, \hat{M}) - J(\pi_\beta, M)| \leq \left(\frac{C_{r,\delta}}{1-\gamma} + \frac{\gamma R_{max} C_{T,\delta}}{(1-\gamma)^2}\right) \mathbb{E}_{s \sim d_{\hat{M}}^{\pi^*}(s)}\left[\frac{\sqrt{|\mathcal{A}|}}{\sqrt{|\mathcal{D}(s)|}}\sqrt{E_{a \sim \pi^*(a|s)}\left(\frac{\pi^*(a|s)}{\pi_\beta(a|s)}\right)}\right]$$

$$(32)$$

given that $\sqrt{E_{a \sim \pi^*(a|s)}\left[\frac{\pi^*(a|s)}{\pi_\beta(a|s)}\right]}$ is a pointwise upper bound of $\sqrt{E_{a \sim \pi_\beta(a|s)}\left[\frac{\pi_\beta(a|s)}{\pi_\beta(a|s)}\right]}$([4]). Thus we get,

$$J(\pi^*, \hat{M}) \geq J(\pi_\beta, \hat{M}) - 2\left(\frac{C_{r,\delta}}{1-\gamma} + \frac{\gamma R_{max} C_{T,\delta}}{(1-\gamma)^2}\right) \mathbb{E}_{s \sim d_{\hat{M}}^{\pi^*}(s)}\left[\frac{\sqrt{|\mathcal{A}|}}{\sqrt{|\mathcal{D}(s)|}}\sqrt{E_{a \sim \pi^*(a|s)}\left(\frac{\pi^*(a|s)}{\pi_\beta(a|s)}\right)}\right]$$

$$+ \alpha \frac{1}{1-\gamma} \mathbb{E}_{s \sim d_{\hat{M}}^\pi(s)}\left[\frac{d(s)}{d_u(s)} - 1\right]$$

$$(33)$$

which completes the proof. $\qquad \square$

Here, the second term represents the sampling error, which arises due to the discrepancy between $\hat{M}$ and $M$. The third term signifies the enhancement in policy performance attributed to our algorithm in $\hat{M}$. It's worth noting that when the first term is minimized, smaller values of $\alpha$ can achieve improvements over the behavior policy.

## B   CSVE Algorithm and Implementation Details

In Section 4, we deteiled the complete formula descriptions of the CSVE practical offline RL algorithm. Here we consolidate those details and present the full deep offline reinforcement learning algorithm as illustrated in Alg. 1. In particular, the dynamic model, value functions, and policy are parameterized with deep neural networks and optimized via stochastic gradient descent methods.

We implement our method based on an offline deep reinforcement learning library d3rlpy [34]. The code is available at: https://github.com/2023AnnonymousAuthor/csve .

Table 3: Hyper-parameters of CSVE evaluation

| Hyper-parameters | Value and description |
|---|---|
| B | 5, number of ensembles in dynamics model |
| $\alpha$ | 10, to control the penalty of OOD states |
| $\tau$ | 10, budget parameter in Eq. 8 |
| $\beta$ | In Gym domain, 3 for random and medium tasks, 0.1 for the other tasks; In Adroit domain, 30 for human and cloned tasks, 0.01 for expert tasks |
| $\gamma$ | 0.99, discount factor. |
| H | 1 million for Mujoco while 0.1 million for Adroit tasks. |
| w | 0.005, target network smoothing coefficient. |
| lr of actor | 3e-4, policy learning rate |
| lr of critic | 1e-4, critic learning rate |

## C  Additional Ablation Study

**Effect of model errors.** Compared to traditional model-based offline RL algorithms, CSVE exhibits greater resilience to model biases. To access this resilience quantitatively, we measured the performance impact of model biases using the average L2 error in transition prediction as an indicator. As shown in Fig. 3, the influence of model bias on RL performance is CSVE is marginal. Specifically, in the halfcheetah task, there is no observable impact of model errors on scores, model errors show no discernible impact on scores. For the hopper and walker2d tasks, only a minor decline in scores is observed as the errors escalate.

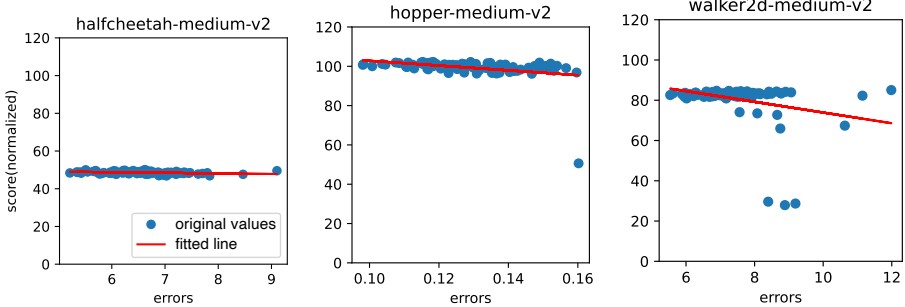

Figure 3: Effect of the model biases to performance scores. The correlation coefficient is $-0.32$, $-0.34$, and $-0.29$ respectively.

## D  Experimental Details and Complementary Results

### D.1  Hyper-parameters of CSVE evaluation in experiments

Table 3 provides a detailed breakdown of the hyper-parameters used for evaluating CSVE in our experiments.

### D.2  Details of Baseline CQL-AWR

To facilitate a direct comparison between the effects of conservative state value estimation and Q-value estimation, we formulated a baseline method named CQL-AWR as detailed below:

$$\hat{Q}^{k+1} \leftarrow \arg\min_Q \alpha \left( E_{s \sim D, a \sim \pi(a|s)}[Q(s,a)] - E_{s \sim D, a \sim \hat{\pi}_\beta(a|s)}[Q(s,a)] \right) + \frac{1}{2} E_{s,a,s' \sim D}[(Q(s,a) - \hat{\beta}_\pi \hat{Q}^k(s,a))^2]$$

$$\pi \leftarrow \arg\min_{\pi'} L_\pi(\pi') = -E_{s,a \sim D}\left[ \log \pi'(a|s) \exp\left( \beta \hat{A}^{k+1}(s,a) \right) \right] - \lambda E_{s \sim D, a \sim \pi'(s)}\left[ \hat{Q}^{k+1}(s,a) \right]$$

$$\text{where } \hat{A}^{k+1}(s,a) = \hat{Q}^{k+1}(s,a) - \mathbb{E}_{a \sim \pi}[\hat{Q}^{k+1}(s,a)].$$

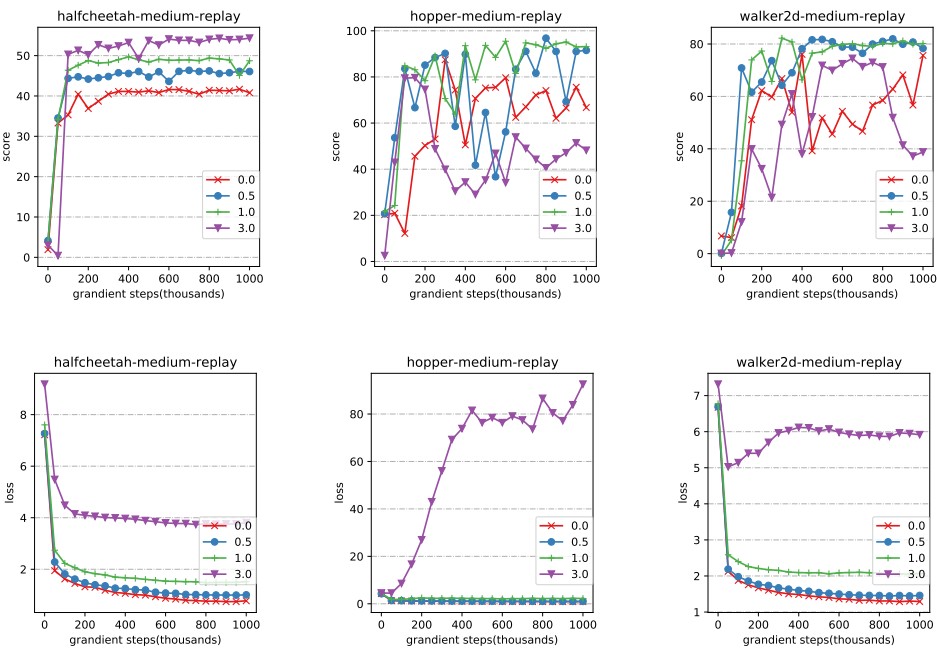

Figure 4: Effect of $\lambda$ to Score (upper figures) and $L_\pi$ loss in Eq. 9 (bottom figures)

In CQL-AWR, the critic adopts a standard CQL equation, while the policy improvement part uses an AWR extension combined with novel action exploration as denoted by the conservative Q function. When juxtaposed with our CSVE implementation, the policy segment of CQL-AWR mirrors ours, with the primary distinction being that its exploration is rooted in a Q-based and model-free approach.

## D.3 Reproduction of COMBO

In Table 1 of our main paper, we used the results of COMBO as presented in the literature [23]. Here we detail additional attempts to reproduce the results and compare the performance of CSVE with COMBO.

**Official Code.** We initially aimed to rerun the experiment using the official COMBO code provided by the authors. The code is implemented in Tensoflow 1.x and relies on software versions from 2018. Despite our best efforts to recreate the computational environment, we encountered challenges in reproducing the reported results. For instance, Fig. 5 illustrates the asymptotic performance on medium datasets up to 1000 epochs, where the scores have been normalized based on SAC performance metrics. Notably, for both the hopper and walker2d tasks, the performance scores exhibited significant variability. The average scores over the last 10 epochs for halfcheetah, hopper, and walker2d were 71.7, 65.3, and -0.26, respectively. Furthermore, we observed that even when using the D4RL v0 dataset, COMBO demonstrated similar performance patterns when recommended hyper-parameters were applied.

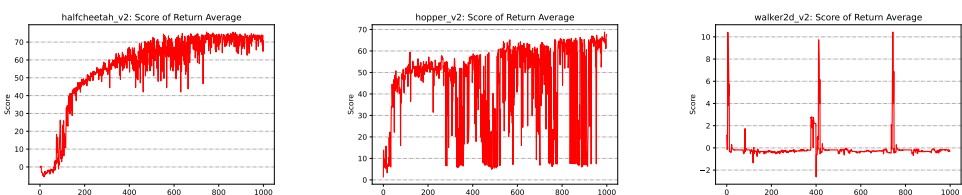

Figure 5: Return of official COMBO implementation on D4RL mujoco v2 tasks, fixing seed=0.

**JAX-based optimized implementation Code [35].** We also tested one recent re-implementation available in RIQL. This version is regarded as the most highly-tuned implementation to date. The results of our tests can be found in Fig.6. For the random and expert datasets, we applied the same hyper-parameters as those used for the medium and medium-expert datasets, respectively. For all other datasets, we adhered to the default hyper-parameters provided by the authors [35]. Despite these efforts, when we compared our outcomes with the original authors' results (as shown in Table 10 and Fig.7 of [35]), our reproduced results consistently exhibited both lower performance scores and greater variability.

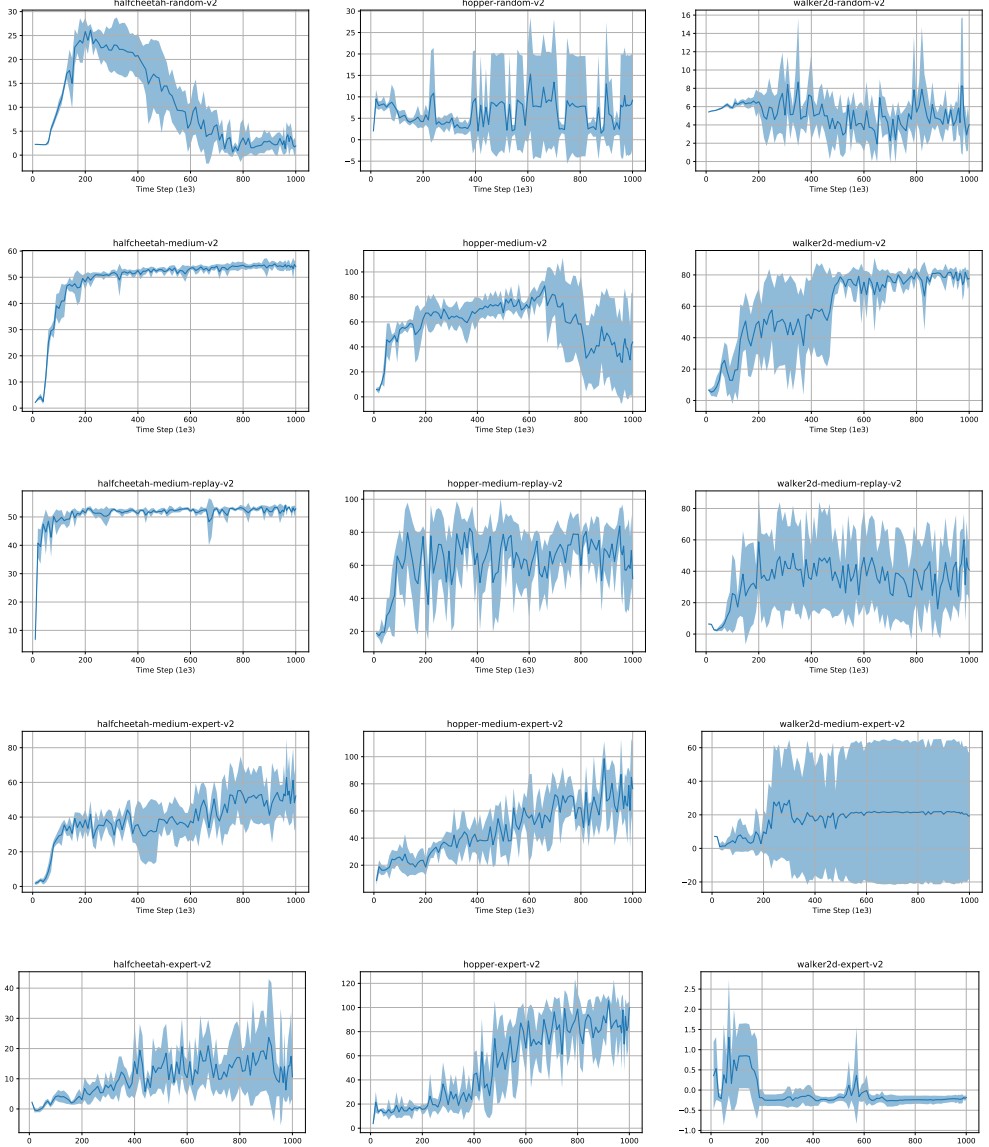

Figure 6: Return of an optimized COMBO implementation[35] on D4RL mujoco v2 tasks. The data are obtained by running with 5 seeds for each task, and the dynamics model has 7 ensembles.

## D.4 Effect of Exploration on Near Dataset Distributions

As discussed in Section 3.1 and 4.2, the appropriate selection of exploration on the distribution ($d$) beyond data ($d_u$) should help policy improvement. The factor $\lambda$ in Eq. 10 controls the trade-off on such 'bonus' exploration and complying with the data-implied behavior policy.

In section 5.2, we examined the effect of $\lambda$ on the medium datasets of mujoco tasks. Now let us further take the medium-replay type of datasets for more analysis of its effect. In the experiments, with fixed $\beta = 0.1$, we investigate $\lambda$ values of $\{0.0, 0.5, 1.0, 3.0\}$. As shown in the upper figures in Fig. 4, $\lambda$ shows an obvious effect on policy performance and variances during training. In general, there is a value under which increasing $\lambda$ leads to performance improvement, while above which further increasing $\lambda$ hurts performance. For example, with $\lambda = 3.0$ in hopper-medium-replay task and walker2d-medium-replay task, the performance gets worse than with smaller $\lambda$ values. The value of $\lambda$ is task-specific, and we find that its effect is highly related to the loss in Eq. 9 which can be observed by comparing the bottom and upper figures in Fig. 4. Thus, in practice, we can choose proper $\lambda$ according to the above loss without online interaction.

### D.5 Conservative State Value Estimation by Perturbing Data State with Noise

In this section, we investigate a model-free method for sampling OOD states and compare its results with the model-based method adopted in section 4.

The model-free method samples OOD states by randomly adding Gaussian noise to the sampled states from the data. Specifically, we replace the Eq.5 with the following Eq. 34, and other parts are consistent with the previous technology.

$$
\begin{aligned}
\hat{V}^{k+1} \leftarrow \arg\min_V L_V^\pi(V; \overline{\hat{Q}^k}) = \; & \alpha \left( E_{s\sim D, s'=s+N(0,\sigma^2)}[V(s')] - E_{s\sim D}[V(s)] \right) \\
& + E_{s\sim D}\left[ (E_{a\sim\pi(\cdot|s)}[\overline{\hat{Q}^k}(s,a)] - V(s))^2 \right].
\end{aligned}
\tag{34}
$$

The experimental results on the Mujoco control tasks are summarized in Table 4. As shown, with different noise levels ($\sigma^2$), the model-free CSVE perform worse than our original model-based CSVE implementation; and for some problems, the model-free method shows very large variances across seeds. Intuitively, if the noise level covers the reasonable state distribution around data, its performance is good; otherwise, it misbehaves. Unfortunately, it is hard to find a noise level that is consistent for different tasks or even the same tasks with different seeds.

Table 4: Performance comparison on Gym control tasks. The results of different noise levels are over three seeds.

| | | CQL | CSVE | $\sigma^2$=0.05 | $\sigma^2$=0.1 | $\sigma^2$=0.15 |
|---|---|---|---|---|---|---|
| Random | HalfCheetah | $17.5 \pm 1.5$ | $26.7 \pm 2.0$ | $20.8 \pm 0.4$ | $20.4 \pm 1.3$ | $18.6 \pm 1.1$ |
| | Hopper | $7.9 \pm 0.4$ | $27.0 \pm 8.5$ | $4.5 \pm 3.1$ | $14.2 \pm 15.3$ | $6.7 \pm 5.4$ |
| | Walker2D | $5.1 \pm 1.3$ | $6.1 \pm 0.8$ | $3.9 \pm 3.8$ | $7.5 \pm 6.9$ | $1.7 \pm 3.5$ |
| Medium | HalfCheetah | $47.0 \pm 0.5$ | $48.6 \pm 0.0$ | $48.2 \pm 0.2$ | $47.5 \pm 0.0$ | $46.0 \pm 0.9$ |
| | Hopper | $53.0 \pm 28.5$ | $99.4 \pm 5.3$ | $36.9 \pm 32.6$ | $46.1 \pm 2.1$ | $18.4 \pm 30.6$ |
| | Walker2D | $73.3 \pm 17.7$ | $82.5 \pm 1.5$ | $81.5 \pm 1.0$ | $75.5 \pm 1.9$ | $78.6 \pm 2,9$ |
| Medium Replay | HalfCheetah | $45.5 \pm 0.7$ | $54.8 \pm 0.8$ | $44.8 \pm 0.4$ | $44.1 \pm 0.5$ | $43.8 \pm 0.4$ |
| | Hopper | $88.7 \pm 12.9$ | $91.7 \pm 0.3$ | $85.5 \pm 3.0$ | $78.3 \pm 4.3$ | $70.2 \pm 12.0$ |
| | Walker2D | $81.8 \pm 2.7$ | $78.5 \pm 1.8$ | $78.7 \pm 3.3$ | $76.8 \pm 1.3$ | $66.8 \pm 4.0$ |
| Medium Expert | HalfCheetah | $75.6 \pm 25.7$ | $93.1 \pm 0.3$ | $87.5 \pm 6.0$ | $89.7 \pm 6.6$ | $93.8 \pm 1.6$ |
| | Hopper | $105.6 \pm 12.9$ | $95.2 \pm 3.8$ | $63.2 \pm 54.4$ | $99.0 \pm 11.0$ | $37.6 \pm 63.9$ |
| | Walker2D | $107.9 \pm 1.6$ | $109.0 \pm 0.1$ | $108.4 \pm 1.9$ | $109.5 \pm 1.3$ | $110.4 \pm 0.6$ |
| Expert | HalfCheetah | $96.3 \pm 1.3$ | $93.8 \pm 0.1$ | $59.0 \pm 28.6$ | $67.5 \pm 21.9$ | $75.3 \pm 27.3$ |
| | Hopper | $96.5 \pm 28.0$ | $111.2 \pm 0.6$ | $67.3 \pm 57.7$ | $109.2 \pm 2.4$ | $109.4 \pm 2.1$ |
| | Walker2D | $108.5 \pm 0.5$ | $108.5 \pm 0.0$ | $109.7 \pm 1.1$ | $108.9 \pm 1.6$ | $108.6 \pm 0.3$ |

