# A  Proofs

439  We first redefine notation for clarity and then provide the proofs of the results in the main paper.

440  **Notation**. Let $k \in N$ denote an iteration of policy evaluation(in Section 3.2). $V^k$ denotes the
441  true, tabular (or functional) V-function iterate in the MDP, without any correction. $\hat{V}^k$ denotes the
442  approximate tabular (or functional) V-function iterate.

443  The empirical Bellman operator can be expressed as follows:

$$(\hat{\mathcal{B}}^\pi \hat{V}^k)(s) = E_{a\sim\pi(a|s)}\hat{r}(s,a) + \gamma \sum_{s'} E_{a\sim\pi(a|s)}\hat{P}(s'|s,a)[\hat{V}^k(s')] \tag{10}$$

444  where $\hat{r}(s,a)$ is the empirical average reward obtained in the dataset when performing action $a$ at
445  state $s$. The true Bellman operator can be expressed as follows:

$$(\mathcal{B}^\pi V^k)(s) = E_{a\sim\pi(a|s)}r(s,a) + \gamma \sum_{s'} E_{a\sim\pi(a|s)}P(s'|s,a)[V^k(s')] \tag{11}$$

446  Now we first prove that the iteration in Eq.2 has a fixed point. Assume state value function is lower
447  bounded, i.e., $V(s) \geq C \ \forall s \in S$, then Eq.2 can always be solved with Eq.3. Thus, we only need to
448  investigate the iteration in Eq.3.

449  Denote the iteration as a function operator $\mathcal{T}^\pi$ on $V$. Suppose $\text{supp}\,d \subseteq \text{supp}\,d_u$, then the operator
450  $\mathcal{T}^\pi$ is a $\gamma$-contraction in $L_\infty$ norm where $\gamma$ is the discounting factor.

451  **Proof of Lemma 3.1:** Let $V$ and $V'$ are any two state value functions with the same support, i.e.,
452  $\text{supp}V = \text{supp}V'$.

$$|(\mathcal{T}^\pi V - \mathcal{T}^\pi V')(s)| = \left|(\hat{\mathcal{B}}^\pi V(s) - \alpha[\frac{d_{(s)}}{d_u(s)} - 1]) - (\hat{\mathcal{B}}^\pi V'(s) - \alpha[\frac{d_{(s)}}{d_u(s)} - 1])\right|$$

$$= \left|\hat{\mathcal{B}}^\pi V(s) - \hat{\mathcal{B}}^\pi V'(s)\right|$$

$$= |(E_{a\sim\pi(a|s)}\hat{r}(s,a) + \gamma E_{a\sim\pi(a|s)}\sum_{s'}\hat{P}(s'|s,a)V(s'))$$

$$- (E_{a\sim\pi(a|s)}\hat{r}(s,a) + \gamma E_{a\sim\pi(a|s)}\sum_{s'}\hat{P}(s'|s,a)V'(s'))|$$

$$= \gamma\left|E_{a\sim\pi(a|s)}\sum_{s'}\hat{P}(s'|s,a)[V(s') - V'(s')]\right|$$

$$||\mathcal{T}^\pi V - \mathcal{T}^\pi V'||_\infty = \max_s |(\mathcal{T}^\pi V - \mathcal{T}^\pi V')(s)|$$

$$= \max_s \gamma\left|E_{a\sim\pi(a|s)}\sum_{s'}\hat{P}(s'|s,a)[V(s') - V'(s')]\right|$$

$$\leq \gamma E_{a\sim\pi(a|s)}\sum_{s'}\hat{P}(s'|s,a)\max_{s''}|V(s'') - V'(s'')|$$

$$= \gamma\max_{s''}|V(s'') - V'(s'')|$$

$$= \gamma||(V - V')||_\infty$$

453 $\qquad\qquad\qquad\qquad\qquad\qquad\qquad\qquad\qquad\qquad\qquad\qquad\qquad\qquad\qquad\qquad\qquad\qquad$ $\square$

454  We present the bound on using empirical Bellman operator compared to the true Bellman operator.
455  Following previous work [4], we make the following assumptions that: $P^\pi$ is the transition matrix
456  coupled with policy, specifically, $P^\pi V(s) = E_{a'\sim\pi(a'|s'),s'\sim P(s'|s,a')}[V(s')]$

457  **Assumption A.1.** $\forall s, a \in \mathcal{M}$, the following relationships hold with at least $(1-\delta)$ $(\delta \in (0,1))$
458  probability,

$$|r - r(s,a)| \leq \frac{C_{r,\delta}}{\sqrt{|D(s,a)|}}, ||\hat{P}(s'|s,a) - P(s'|s,a)||_1 \leq \frac{C_{P,\delta}}{\sqrt{|D(s,a)|}} \tag{12}$$

Under this assumption, the absolute difference between the empirical Bellman operator and the actual one can be calculated as follows:

$$|(\hat{\mathcal{B}}^\pi)\hat{V}^k - (\mathcal{B}^\pi)\hat{V}^k)| = E_{a\sim\pi(a|s)}|r - r(s,a) + \gamma\sum_{s'} E_{a'\sim\pi(a'|s')}(\hat{P}(s'|s,a) - P(s'|s,a))[\hat{V}^k(s')]| \tag{13}$$

$$\leq E_{a\sim\pi(a|s)}|r - r(s,a)| + \gamma|\sum_{s'} E_{a'\sim\pi(a'|s')}(\hat{P}(s'|s,a') - P(s'|s,a'))[\hat{V}^k(s')]| \tag{14}$$

$$\leq E_{a\sim\pi(a|s)}\frac{C_{r,\delta} + \gamma C_{P,\delta}2R_{max}/(1-\gamma)}{\sqrt{|D(s,a)|}} \tag{15}$$

Thus, the estimation error due to sampling error can be bounded by a constant as a function of $

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

$$(32)$$

which completes the proof. $\square$

Here, the second term is sampling error which occurs due to mismatch of $\hat{M}$ and $M$; the third term denotes the increase in policy performance due to CSVE in $\hat{M}$. Note that when the first term is small, the smaller value of $\alpha$ are able to provide an improvement compared to the behavior policy.

## B CSVE Algorithm and Implementation Details

In section 4, we have given the complete formula descriptions of a practical offline RL algorithm of CSVE. Here we put all together and describe the practical deep offline reinforcement learning algorithm in Alg. 1. In particular, the dynamic model model, value functions and policy are all parameterized with deep neural networks and trained via stochastic gradient decent methods.

We implement our method based on an offline deep reinforcement learning library d3rlpy [33]. The code is available at: https://github.com/2023AnnonymousAuthor/csve .

### B.1 Additional ablation study

**Effect of exploration on near states.** We analyze the impact of varying the factor $\lambda$ in Eq. 9, which controls the intensity on such exploration. We investigated $\lambda$ values of $\{0.0, 0.1, 0.5, 1.0\}$ in the

Table 3: Hyper-parameters of CSVE evaluation

| Hyper-parameters | Value and description |
|---|---|
| B | 5, number of ensembles in dynamics model |
| $\alpha$ | 10, to control the penalty of out-of-distribution states |
| $\tau$ | 10, budget parameter in Eq. 8 |
| $\beta$ | In Gym domain, 3 for random and medium tasks, 0.1 for the other tasks; In Adroit domain, 30 for human and cloned tasks, 0.01 for expert tasks |
| $\gamma$ | 0.99, discount factor. |
| $H$ | 1 million for Mujoco while 0.1 million for Adroit tasks. |
| $w$ | 0.005, target network smoothing coefficient. |
| lr of actor | 3e-4, policy learning rate |
| lr of critic | 1e-4, critic learning rate |

medium tasks, fixing $\beta = 0.1$. The results are plotted in Fig. 2. As shown in the upper figures, $\lambda$ has obvious effect to policy performance and variances during training. With increasing $\lambda$ from 0, the converged performance gets better in general. However, when the $\lambda$ becomes too large (e.g., $\lambda = 3$ for hopper and walker2d), the performance may degrade or even collapse. By further investigating the $L_\pi$ loss in Eq.9, as shown in the bottom figures, we found that larger $\lambda$ values have negative effect to $L_\pi$; however, once $L_\pi$ converges low reasonably, the bigger $\lambda$ leads to performance improvement.

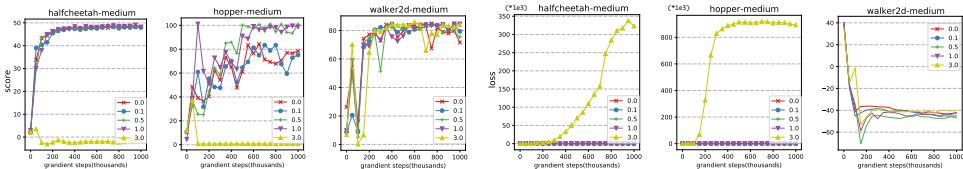

Figure 2: Effect of $\lambda$ to performance scores (upper figures) and losses (bottom figures) in Eq. 9 on medium tasks.

**Effect of model errors.** Compared to traditional model-based offline RL algorithms, CSVE is less affected by model biases. To measure this quantitatively, we studied the impact of model biases on performance by using the average L2 error on transition prediction as a surrogate for model biases. As shown in Fig. 3, the effect of model bias on RL performance is subtle in CSVE. Specifically, for the halfcheetah task, there is no observable impact of model errors on scores, while in the hopper and walker2d tasks, there is only a slight decrease in scores as the errors increase.

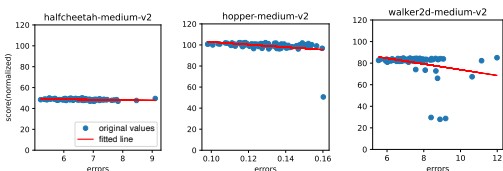

Figure 3: Effect of the model biases to performance scores. The correlation coefficient is $-0.32$, $-0.34$, and $-0.29$ respectively.

# C Experimental Details and Complementary Results

## C.1 Hyper-parameters of CSVE evaluation in experiments

The detailed hyper-parameters of CSVE used in experiments are provided in Table 3.

## C.2 More experiments on hyper-parameters effect

We also investigated $\lambda$ values of $\{0.0, 0.5, 1.0, 3.0\}$ in the medium-replay tasks. The results are shown in Fig. 2.

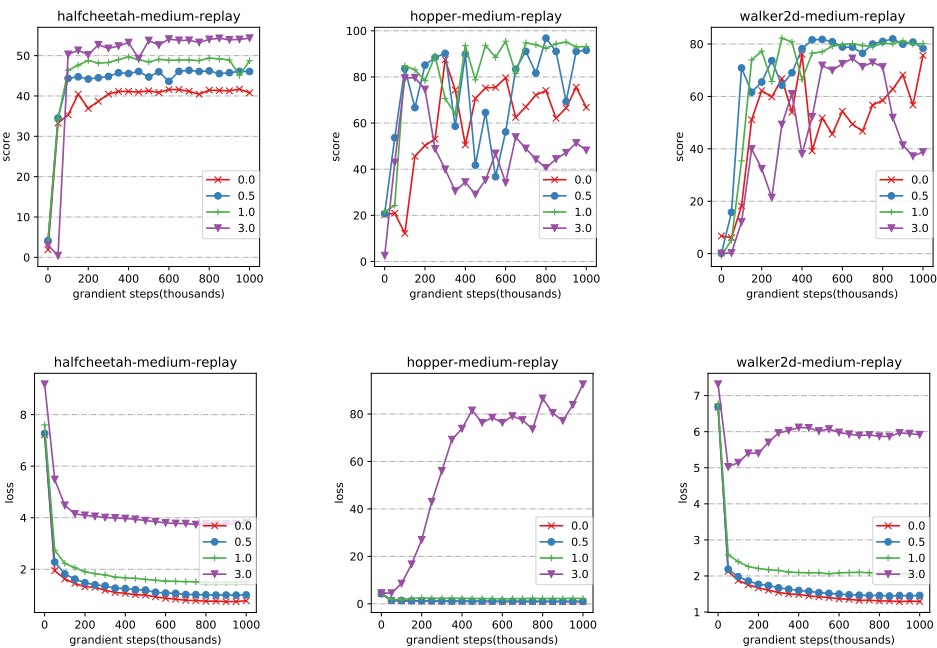

Figure 4: Effect of $\lambda$ to Score (upper figures) and $L_\pi$ loss in Eq. 9 (bottom figures)

## C.3 Details of Baseline CQL-AWR

In order to directly compare effect of the conservative state value estimation against Q value estimation, we implement a baseline method namely CQL-AWR as follows:

$$\hat{Q}^{k+1} \leftarrow \arg\min_Q \alpha \left(E_{s\sim D, a\sim\pi(a|s)}[Q(s,a)] - E_{s\sim D, a\sim\hat{\pi}_\beta(a|s)}[Q(s,a)]\right) + \frac{1}{2}E_{s,a,s'\sim D}[(Q(s,a) - \hat{\beta}_\pi\hat{Q}^k(s,a))^2]$$

$$\pi \leftarrow \arg\min_{\pi'} L_\pi(\pi') = -E_{s,a\sim D}\left[\log\pi'(a|s)\exp\left(\beta\hat{A}^{k+1}(s,a)\right)\right] - \lambda E_{s\sim D, a\sim\pi'(s)}\left[\hat{Q}^{k+1}(s,a)\right]$$

$$\text{where } \hat{A}^{k+1}(s,a) = \hat{Q}^{k+1}(s,a) - \mathbb{E}_{a\sim\pi}[\hat{Q}^{k+1}(s,a)].$$

In CQL-AWR, the critic adopts a normal CQL equation, while the policy improvement part uses a AWR extended with new action exploration indicated by the conservative Q function. Compared to our CSVE implementation, its policy part is similar except that the exploration is Q-based and model-free.

## C.4 Reproduction of COMBO

In Table 1 of our main paper, our results of COMBO adopt the one presented in literature [23]. Here we list other reproducing efforts and results which may be helpful for readers to compare CSVE with COMBO.

**Official Code.** We preferred to rerun the official COMBO code provided by authors. The code is implemented in Tensoflow 1.x and depends on software versions in 2018. We rebuilt the environment but still failed to reproduce the results. For example, Fig. 5 shows the asymptotic performance on medium datasets until 1000 epochs, in which the scores have been normalized with corresponding SAC performance. We found that in both hopper and walker2d, the scores show dramatic fluctuations. The average scores of last 10 epochs for halfcheetah, hopper and walker2d are 71.7, 65.3 and -0.26 in respect. Besides, we found that even in D4RL v0 dataset, COMBO's behaviours are similar with recommended hyper-parameters.

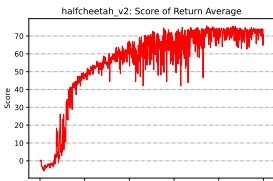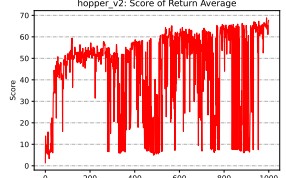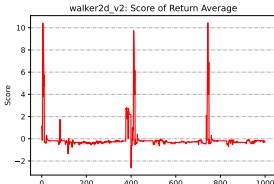

Figure 5: Return of official COMBO implementation on D4RL mujoco v2 tasks, fixing seed=0.

**JAX-based optimized implementation Code [34].** We also rerun one recent re-implementation
in RIQL which is the most highly tuned implementation so far. The results are shown in Fig.6. For
random and expert datasets, we used the same hyper-parameters same with medium and medium-
expert respectively. For all other datasets, we used the default hyper-parameters given by authors
[34]. By comparing with the authors' results (Table 10 and Fig.7 in [34]), our reproduced results are
still lower and with larger variances.

## C.5 Effect of Exploration on near Dataset Distributions

As discussed in Section 3.1 and 4.2, proper choice of exploration on the distribution ($d$) beyond data
($d_u$) should help policy improvement. The factor $\lambda$ in Eq. 9 controls the trade-off on such 'bonus'
exploration and complying the data-implied behaviour policy.

In section 5.2, we have investigated the effect of $\lambda$ in medium datasets of mujoco tasks. Now let us
further take the medium-replay type of datasets for more analysis of its effect. In the experiments,
with fixed $\beta = 0.1$, we investigate $\lambda$ values of $\{0.0, 0.5, 1.0, 3.0\}$. As shown in the upper figures
in Fig. 4, $\lambda$ shows obvious effect to policy performance and variances during training. In general,
there is a value under which increasing $\lambda$ leads to performance improvement, while above which
further increasing $\lambda$ hurts performance. For example, with $\lambda = 3.0$ in hopper-medium-replay task
and walker2d-medium-replay task, the performance get worse than with smaller $\lambda$ values. The value
of $\lambda$ is task-specific, and we find that its effect is highly related to the loss in Eq. 9 which can be
observed by comparing bottom and upper figures in Fig. 4. Thus, in practice, we can choose proper $\lambda$
according to the above loss without online interaction.

## C.6 Conservative State Value Estimation by Perturbing Data State with Noise

In this section, we investigate a model-free method for sampling OOD states, and compare its results
with the model-based method adopted in our implementation in section 4.

The model-free method samples OOD states by randomly adding Gaussian noise to the sampled
states from data. Specifically, we replace the Eq.5 with the following Eq. 33, and other parts are same
as previous.

$$
\hat{V}^{k+1} \leftarrow \arg\min_V L_V^\pi(V; \overline{\hat{Q}^k}) = \alpha \left( E_{s \sim D, s' = s + N(0,\sigma^2)}[V(s')] - E_{s \sim D}[V(s)] \right)
$$
$$
+ E_{s \sim D} \left[ (E_{a \sim \pi(\cdot|s)}[\overline{\hat{Q}^k}(s, a)] - V(s))^2 \right]. \tag{33}
$$

The experimental results on mujoco control tasks are summarized in Table 4. As shown, with different
noise levels ($\sigma^2$), the model-free CSVE may performs better or worse than our original model-based
CSVE implementation; and for some problems, the model-free method show very large variances
across seeds. Intuitively, if the noise level covers the reasonable state distribution around data, its
performance is good; otherwise, it misbehaves. Unfortunately, it is hard to find a noise level that is
consistent for different tasks or even the same tasks with different seeds.

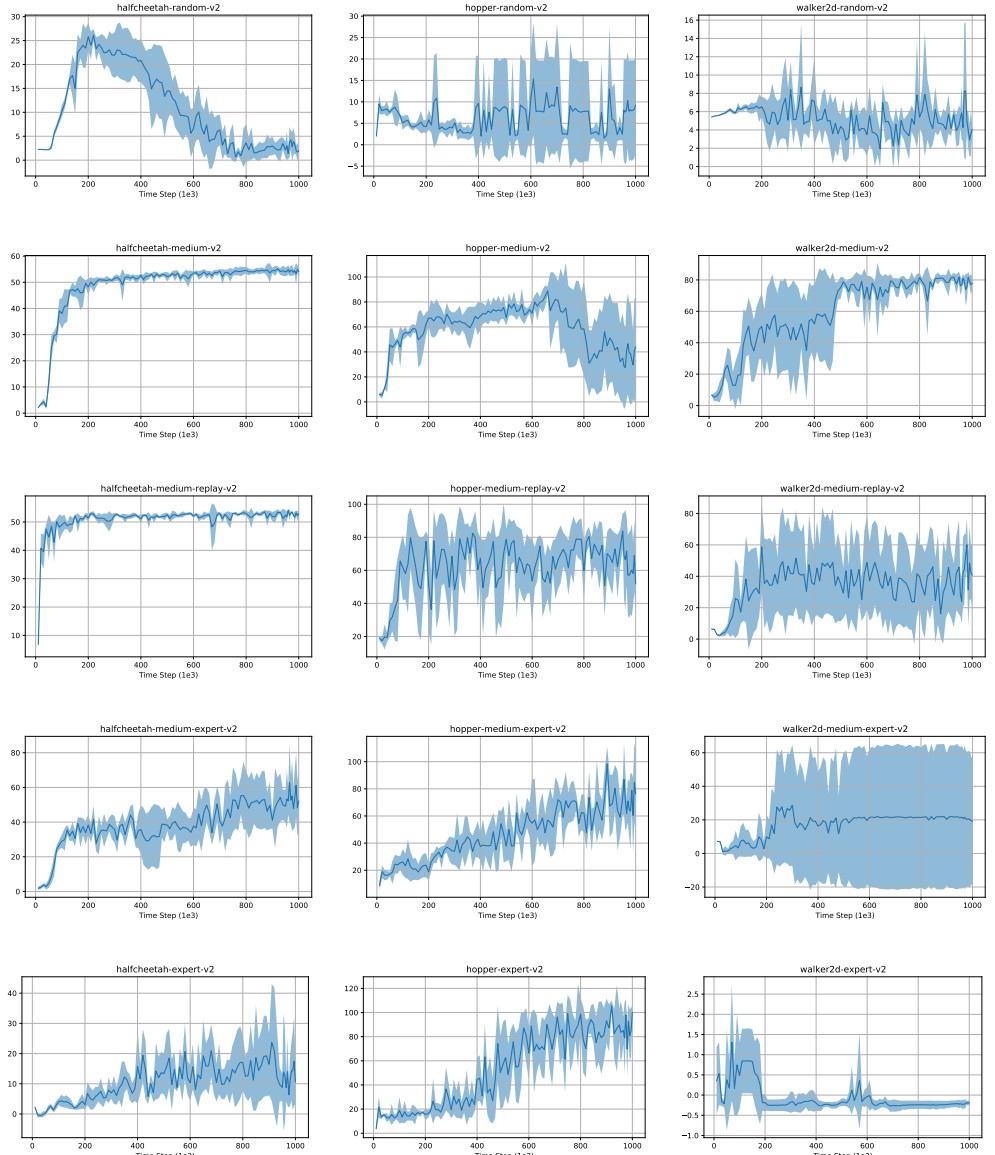

Figure 6: Return of an optimized COMBO implementation[34] on D4RL mujoco v2 tasks. The data are got by running with 5 seeds for each task, and the dynamics model has 7 ensembles.

Table 4: Performance comparison on Gym control tasks. The results of different noise levels is over three seeds.

| | | CQL | CSVE | $\sigma^2$=0.05 | $\sigma^2$=0.1 | $\sigma^2$=0.15 |
|---|---|---|---|---|---|---|
| Random | HalfCheetah | $17.5 \pm 1.5$ | $26.7 \pm 2.0$ | $20.8 \pm 0.4$ | $20.4 \pm 1.3$ | $18.6 \pm 1.1$ |
| | Hopper | $7.9 \pm 0.4$ | $27.0 \pm 8.5$ | $4.5 \pm 3.1$ | $14.2 \pm 15.3$ | $6.7 \pm 5.4$ |
| | Walker2D | $5.1 \pm 1.3$ | $6.1 \pm 0.8$ | $3.9 \pm 3.8$ | $7.5 \pm 6.9$ | $1.7 \pm 3.5$ |
| Medium | HalfCheetah | $47.0 \pm 0.5$ | $48.6 \pm 0.0$ | $48.2 \pm 0.2$ | $47.5 \pm 0.0$ | $46.0 \pm 0.9$ |
| | Hopper | $53.0 \pm 28.5$ | $99.4 \pm 5.3$ | $36.9 \pm 32.6$ | $46.1 \pm 2.1$ | $18.4 \pm 30.6$ |
| | Walker2D | $73.3 \pm 17.7$ | $82.5 \pm 1.5$ | $81.5 \pm 1.0$ | $75.5 \pm 1.9$ | $78.6 \pm 2, 9$ |
| Medium Replay | HalfCheetah | $45.5 \pm 0.7$ | $54.8 \pm 0.8$ | $44.8 \pm 0.4$ | $44.1 \pm 0.5$ | $43.8 \pm 0.4$ |
| | Hopper | $88.7 \pm 12.9$ | $91.7 \pm 0.3$ | $85.5 \pm 3.0$ | $78.3 \pm 4.3$ | $70.2 \pm 12.0$ |
| | Walker2D | $81.8 \pm 2.7$ | $78.5 \pm 1.8$ | $78.7 \pm 3.3$ | $76.8 \pm 1.3$ | $66.8 \pm 4.0$ |
| Medium Expert | HalfCheetah | $75.6 \pm 25.7$ | $93.1 \pm 0.3$ | $87.5 \pm 6.0$ | $89.7 \pm 6.6$ | $93.8 \pm 1.6$ |
| | Hopper | $105.6 \pm 12.9$ | $95.2 \pm 3.8$ | $63.2 \pm 54.4$ | $99.0 \pm 11.0$ | $37.6 \pm 63.9$ |
| | Walker2D | $107.9 \pm 1.6$ | $109.0 \pm 0.1$ | $108.4 \pm 1.9$ | $109.5 \pm 1.3$ | $110.4 \pm 0.6$ |
| Expert | HalfCheetah | $96.3 \pm 1.3$ | $93.8 \pm 0.1$ | $59.0 \pm 28.6$ | $67.5 \pm 21.9$ | $75.3 \pm 27.3$ |
| | Hopper | $96.5 \pm 28.0$ | $111.2 \pm 0.6$ | $67.3 \pm 57.7$ | $109.2 \pm 2.4$ | $109.4 \pm 2.1$ |
| | Walker2D | $108.5 \pm 0.5$ | $108.5 \pm 0.0$ | $109.7 \pm 1.1$ | $108.9 \pm 1.6$ | $108.6 \pm 0.3$ |