# OpenReview forum: "Conservative State Value Estimation for Offline Reinforcement Learning"
_NeurIPS.cc/2023/Conference — NeurIPS 2023 poster_

### Official Review · Reviewer_Qqxx · 2023-07-03

**Soundness:** 1 poor
**Presentation:** 1 poor
**Contribution:** 1 poor
**Rating:** 3
**Confidence:** 4

**Summary:**

This paper proposes Conservative State Value Estimation (CSVE) for offline reinforcement learning, which directly penalizes the V-function on out-of-distribution (OOD) states and guarantees conservative value estimation under specific state distributions. The authors develop a practical actor-critic algorithm based on CSVE and evaluate its performance on classic continual control tasks.

**Strengths:**

The paper has both theoretical and empirical results.

**Weaknesses:**

The significance of the proposed method is not clear. To be more specific, compared with the penalization of Q-function for OOD actions of CQL, the advantage of penalizing the V-function for OOD states is not clear. This penalization alone cannot address the core issue of offline RL - overestimation and extrapolation error. Additional policy constraint needs to be incorporated in the proposed method. Direct penalization of the V-function in offline RL does not affect the Q-value of OOD actions (see Eq. 5 and 6). It is Q used in action selection rather than V, so the agent will still choose over-estimated OOD actions.

The significance of the theories is not clear. The assumption $\text{supp}~ d \subseteq \text{supp}~ d_u$ is very strong and hard to satisfy. Besides, with this assumption satisfied, the algorithm can never penalize the value for OOD states (since $d$ is in-distribution). Theorem 3.2. and Theorem 3.3., that the expected value of the estimated V under $d$ or $d_u$ is a lower bound of the true values, is of little significance. It is still likely that V is severely over-estimated at the states out of $\text{supp}~d_u$.

The paper is not well organized.

The empirical evaluation is not sufficient to support the effectiveness of CSVE's core component - Eq. 5.

**Questions:**

I do not have additional questions.

**Limitations:**

The authors do not discuss about limitations.

---

> ### Author Rebuttal · Authors · 2023-08-10
>
> Thank you very much for the comments. Nevertheless, we believe that there should be some misunderstanding points that require to clarify or discuss further. As commented by other reviewers, the paper does have clear technical contributions and evaluation. We explain on them as bellow.
>
> ## Q1: The significance of CSVE.
>
> > the advantage of penalizing the V-function for OOD states is not clear... This penalization alone cannot address the core issue of offline RL - overestimation and extrapolation error...
>
> Response: It is true that CSVE alone cannot address the OOD actions and derive a conservative policy. However, CSVE does have advantages by indirectly affect the Q-function and policy. By penalizing values for OOD states,  CSVE reduces  excessive exploration in OOD states while allowing reasonable exploration in those states close to the samples. This effect is generated in the second term of Eq. 5, and transferred to the Q-function via Eq.6.
>
> > Direct penalization of the V-function in offline RL does not affect the Q-value of OOD actions (see Eq. 5 and 6). It is Q used in action selection rather than V, so the agent will still choose over-estimated OOD actions.
>
> Response: Our methodology in Section 4.2 should address this concern while preserving the advantage of conservative state value estimation. As demonstrated in Eq. 9, our algorithm selects from in-sample actions and near-sample actions that have higher values even under conservative estimation.
>
> ## Q2: The significance of theory.
>
> > The issue of assumption $supp \ d \subseteq  supp \ d_u$
>
> Response: Let me clarify. In the theory section, the $d_u$ refers to the underlying state distribution with the behaviour policy, and thus its support is not only the samples in the dataset but the whole reachable space of states with the policy $u$. With this setting, it should be reasonable to assume $supp \ d \subseteq  supp \ d_u$. Indeed, prior works including CQL, COMBO and most theoretic papers (see Table 1 in [3]) have the same assumption. This assumption is mainly for convenience in theoretic analysis. In practice, it could be satisfied by constraining the exploration in and near the dataset generated by $d_u$.
>
>
> ## Q3: The empirical evaluation is not sufficient to support the effectiveness of CSVE's core component - Eq. 5.
>
> Response: Directly measuring the effect of the components in Eq.5 is hard. Instead, we evaluated its effectiveness comprehensively through controlled experiments of baselines and alternative components of CSVE. The design logics is as bellow.
> - By comparing with CQL-AWR / IQL / AWAC, we assess the value estimation component of CSVE (Eq.5-6) versus CQL / IQL(expectile) / AWAC (normal TD-based value estimation), under the same policy extraction method AWR.
> - By comparing CSVE and the model-free CSVE in Appendix C.6, we verify the benefits of model-based next state sampling over model-free state perturbation used in Eq.5.
> - Ablation study. In Appendix B.1 and 5.2, we evaluate with varying hyper-parameters $\lambda$ and $\beta$ to assess how the policy extraction is affected by the conservative state value estimation.
>
>
> **Table R1**: Implementation comparision among different offline RL algorithms.
> Algorithm | Evalue estimation | Policy improvement |
> --- |:---:|:---:|
> CQL|Q-values with penality on OOD actions|SAC-style Actor|
> CQL-AWR|Q-values with penality on OOD actions|AWR + State exploration|
> IQL|Q-values with expectile regression|AWR|
> AWAC|Normal Q-values|AWR|
> COMBO|Q-values with penalty on OOD states and actions (via multi-step model rollouts)|SAC-style Actor|
> CSVE (default)|V-values with penalty on OOD states (via 1-step model rollouts) |AWR + State exploration|
> CSVE (Appendix C.6)|V-values with penalty on OOD states (via perturbing in-sample states) |AWR + State exploration|
>
> References:
> [1]CQL; [2]COMBO; [3] Pessimistic Model-based Offline Reinforcement Learning under Partial Coverage, 2022.

---

> > ### Comment · Reviewer_Qqxx · 2023-08-15
> >
> > Thank you for the reply. After reading the rebuttal, I still have the following questions or concerns.
> > ## Q1
> > > The authors mentioned "This effect is generated in the second term of Eq. 5, and transferred to the Q-function via Eq.6."
> >
> > However in Eq. 6 both s and s' are from the dataset. Hence, I think the effect that "CSVE reduces excessive exploration in OOD states" can not be transferred to the Q-function via Eq. 6, since there is no OOD state in Eq. 6.
> >
> > ## Q2
> > I understand $d_u$ is the underlying state distribution with the behaviour policy. OOD states refer to the ones out of the distribution of  $d_u$. It seems the authors have not addressed my concerns:
> >
> > 1. "Besides, with this assumption satisfied, the algorithm can never penalize the value for OOD states, since d is in-distribution."
> >
> > To be more specific, under the condition $supp d \subseteq suppd_u$, Eq. 5 can never penalize the value for OOD states $s \notin supp~d_u$.
> >
> > 2. "Theorem 3.2. and Theorem 3.3., that the expected value of the estimated V under $d$ or $d_u$ is a lower bound of the true values, is of little significance. It is still likely that V is severely over-estimated at the states out of $supp~d_u$."
> >
> > To be more specific, Theorem 3.2 and Theorem 3.3 in this paper only ensure underestimation of $\mathbb{E}_{s\sim d}V$ and
> >
> > $\mathbb{E}_{s \sim d_u} V$. Since both $d$ and $d_u$ are in distribution because of the assumption, the both theorems have no assurance for OOD states, whose value functions need to be underestimated the most. Besides, even in $supp~d_u$, some states may be highly overestimated.

---

> > > ### Author Response · Authors · 2023-08-16
> > >
> > > Thank you for the response. To facilitate further discussion, it is essential for us to have a clear understanding and agreement on the concepts of out-of-support and out-of-distribution, as there might be some confusion surrounding them. Out-of-distribution (OOD) and out-of-support refer to distinct aspects of data and probability distributions: OOD pertains to data points or samples that do not originate from the same underlying distribution as the training data, whereas out-of-support concerns data points or samples that possess a zero probability (or probability density) under a given probability distribution. Under the given assumption, both $d$ and $d_u$ are considered in support, but they do not belong to the same distribution.
> > >
> > >
> > >
> > > ## Q1 follow-up
> > >
> > > > the effect that "CSVE reduces excessive exploration in OOD states" can not be transferred to the Q-function via Eq. 6, since there is no OOD state in Eq. 6.
> > >
> > > To be more precise, the correct statement should be "there is no out-of-sample state in Eq. 6." The $s'$ could still be considered out-of-distribution if it is rarely reached in the trajectory of the dataset. In Eq. 5, since the $s'$ values are sampled using a dynamics model and penalized, their values are underestimated unless they are in or close enough to the dataset. This is due to the maximization of $E_{s\in D} [V]$ and the effect of implicit neural network continuity regularization.
> > >
> > > We acknowledge that Eq.6 is a comprise and practical implementation. In principle, the expectation should be taken as $E_{s\sim D, a \sim \pi, s' \sim \hat{P}}$, which however introduces the predicative reward $\hat{r}(s,a)$ (or $\hat{r}(s, a, s')$ in some tasks) that is hard to handle. Instead, since for $s \in D$ the $(s, a \sim \pi(\cdot|s))$ is almost in or close to D, we use  Eq.6 as an approximatation.
> > >
> > >
> > > ## Q2 follow-up
> > >
> > >
> > >
> > > Given the assumption, both $d$ and $d_u$ are in support but not in the same distribution.
> > >
> > > > Q2 1: under the condition $supp d \subseteq suppd_u$,, Eq. 5 can never penalize the value for OOD states $s \notin supp~d_u$.
> > >
> > > Right, and it is just what we suppose to do.
> > >
> > > > Q2 2: Since both $d$ and $d_u$ are in distribution because of the assumption, the both theorems have no assurance for OOD states, whose value functions need to be underestimated the most. Besides, even in $supp~d_u$, some states may be highly overestimated.
> > >
> > > Under the definition of out-of-distribution and out-of-support, it is incorrect to say 'both $d$ and $d_u$ are in distribution', and 'the both theorems' do have 'assurance for OOD states' (but no such assurance on out-of-support states).

---

> > > > ### Comment · Reviewer_Qqxx · 2023-08-21
> > > >
> > > > Thank you for your reply. However, after reading it, my concern still remains.
> > > >
> > > > > OOD pertains to data points or samples that do not originate from the same underlying distribution as the training data, whereas out-of-support concerns data points or samples that possess a zero probability (or probability density) under a given probability distribution.
> > > >
> > > > Are there offline RL works that have a clear differentiation between these two concepts? I suppose they are almost the same. (It does not matter and is not my concerns.)
> > > >
> > > > > The s' could still be considered out-of-distribution if it is rarely reached in the trajectory of the dataset.
> > > >
> > > > To the best of my knowledege, there are no offline RL work that considers s in the dataset as OOD. Besides, "True" OOD states that are out of the datatset are not in Eq. 6.
> > > >
> > > > > Right, and it is just what we suppose to do.
> > > >
> > > > Why? Isn't CSVE meant to punish the V-function for OOD states and reduce excessive exploration in OOD states?
> > > >
> > > > > Under the definition of out-of-distribution and out-of-support, it is incorrect to say 'both d and $d_u$ are in distribution', and 'the both theorems' do have 'assurance for OOD states' (but no such assurance on out-of-support states).
> > > >
> > > > Under your definition of out-of-distribution and out-of-support, just replace OOD in my response with out-of-support, and my concern still remains. "Since both $d$ and $d_u$ are in support because of the assumption, the both theorems have no assurance for out-of-support states, whose value functions need to be underestimated the most. Besides, even in $d_u$, some states may be highly overestimated."

---

> > > > > ### Author Response · Authors · 2023-08-21
> > > > >
> > > > > Thanks for your feedback. We insist on the definitions that are standard without ambiguities or confusion in machine learning and statistics. And under these definitions, our paper is consistent.
> > > > >
> > > > > > Are there offline RL works that have a clear differentiation between these two concepts?
> > > > >
> > > > > Yes. To the best of our knowledge, all offline RL works that include theoretic analysis do have clear differentiation on these concepts (support and sample; out-of-distribution, out-of-sample and out-of-support). The literatures we are familiar with include CQL, COMBO, the works on Offline Policy Evaluation, the works[3] I concerned above, to list a few. On page 22 of [4], the author presents an example that illustrates the distinction between in-support constraint and in-distribution constraint when applied to policies. This example provides a clear differentiation between these concepts.
> > > > >
> > > > > Besides, in terms of implementation, many works including the expectile trick in IQL and CVAE-based behaviour policy reconstruction in MCQ[5] all implicitly underestimate the $(s,a) \in D$ that occasionally has very high values.
> > > > >
> > > > > > there are no offline RL work that considers s in the dataset as OOD.
> > > > >
> > > > > This statement may not be entirely accurate.
> > > > >
> > > > > As mentioned on Page 10 of [4]:
> > > > > "...,since $d^\pi (s)$ may be very different from $d^{\pi^\beta} (s)$. In these out-of-distribution states, the generalization error bound no longer holds"
> > > > >
> > > > > According to this description, in the context of offline RL, a state 's' is considered out-of-distribution when $d^\pi(s)$ is significantly different from $d^{\pi^\beta}(s)$. This suggests that offline RL works do take into account some s' in the dataset as out-of-distribution even though these states are in-sample.
> > > > >
> > > > >
> > > > >
> > > > > >Besides, "True" OOD states that are out of the dataset are not in Eq. 6.
> > > > >
> > > > > Indeed, given that Eq. 6 is a compromise and a practical implementation, it is possible that the distribution of s' selected using this equation may be significantly different from $d^{\pi^\beta}(s)$, especially when the samples are limited. Consequently, the out-of-dataset (OOD) states may be included in Eq. 6.
> > > > >
> > > > >
> > > > > > Why? Isn't CSVE meant to punish the V-function for OOD states and reduce excessive exploration in OOD states?
> > > > >
> > > > > I suppose the 'OOD' means 'out-of-support', and explain together with the next question.
> > > > >
> > > > > > Under your definition of out-of-distribution and out-of-support, just replace OOD in my response with out-of-support, and my concern still remains...
> > > > >
> > > > > It is true that 'both theorems have no assurance for out-of-support points', but **this is definitely reasonable**. For any distribution $d$, it is not meaningful to perform any work (including value underestimation) on out-of-support points, as these points will not be sampled. From a theoretical perspective, it is convenient to assume a large support space and consider all points (including those not in or far from the dataset) have a probability density of at least $c \gt 0$; here, $c$ is any constant that can be arbitrarily small but larger than 0. From the perspective of implementation, almost all algorithms operate only around the dataset, sampling points from the original dataset (i.e., in-sample learning), or from the perturbed/interpolated dataset (e.g., CQL, COMBO respectively), or from parameterized distribution of the dataset (e.g., through VAE and GAN). It is theoretically true that those points far from the dataset should be under-estimated the most, but we are meant to not explore them since without strong priors on the environment dynamics and groundtruth behaviour policy the offline dataset itself cannot support such work.
> > > > >
> > > > > [4] Offline Reinforcement Learning: Tutorial, Review, and Perspectives on Open Problems, 2020.
> > > > > [5] Mildly Conservative Q-Learning for Offline Reinforcement Learning, 2022.

---

> > > > > > ### Comment · Reviewer_Qqxx · 2023-08-21
> > > > > >
> > > > > > Thanks for your reply.
> > > > > >
> > > > > > >  To the best of our knowledge, all offline RL works that include theoretic analysis do have clear differentiation on these concepts...
> > > > > >
> > > > > > I have read all the works you mentioned. On page 22 of [4], the example is about "support constraint" (constrain the policy to be in the support of behaviour policy) and "distribution constraint" (contrain the distribution of policy and that of the behaviour policy to be similar, like via KL). In offline RL papers, such as the papers you mentioned, I suppose "in-distribution" and "out-of-distribution" generically mean "in-support" and "out-of-support" respectively. In all my responses, "OOD" means this. It is not my concern about this work. Just make definitions consistent between us.
> > > > > >
> > > > > > > It is true that 'both theorems have no assurance for out-of-support points', but this is definitely reasonable...
> > > > > >
> > > > > > According to your response, CSVE can not and is not meant to punish out-of-support states. Then in Eq.9 where CSVE guides the policy, the second term will prefer $\pi$ that leads to out-of-support s' whose value function is over-estimated. That's why I think the CSVE method should punish out-of-support states rather than in-support ones and the theorem should have assurance for out-of-support states.

---

> > > > > > > ### Author Response · Authors · 2023-08-21
> > > > > > >
> > > > > > > Thanks for the quick response. However, if the reviewer really accepted the standard and rigorous definitions of in-/out-of-support and in-/out-of-distribution, we believe, the above last two concerns should have been already addressed in our responses.

---

### Official Review · Reviewer_ze4E · 2023-07-05

**Soundness:** 3 good
**Presentation:** 3 good
**Contribution:** 3 good
**Rating:** 6
**Confidence:** 4

**Summary:**

The paper proposes a method to tackle the overestimation of values in offline RL by focussing on state-values instead of state-action values and using in-data policy optimization techniques based on model-based RL. They propose an actor-critic variation for their approach and apply the method on various offline RL tasks.

**Strengths:**

1. The proposed method is derivative of existing ideas such as conservative Q learning, which I see as a pro because it does not drastically depart from an already well-established algorithm.
2. I think there is appeal in the method in that it gives some sense of how results can be different when we start to incorporate state-based quantities instead of state-action based quantities, and when we include model-based approaches. The paper may be able to spark some interesting ideas for other papers.


**Weaknesses:**

1. The paper should bold the results in their experiments sections. Its incredibly tedious to read and discern where CSVE performs well and where it doesnt.
2. There is little intuition for why state-based methods in this case can work better than state-action. I think given that the tweak is somewhat minimal, the paper should stress why this tweak can actually make things better than is typically done.
3. In terms of bounds, I don’t see a comparison to CQL? That is, there are V-function bounds computed by CSVE and true, but there aren’t V-function bounds computed implicitly by CQL (when CQL computes its Q-functions, and then using that to compute the V-function) and the true V-function. I think that is related to point 2 and it would be illuminating.
4. I find it a bit unsettling that the results of prior work in Table 1 were just copy-pasted here. It's not clear to me if it is an absolutely fair comparison since setups between various papers can be different (random seeds etc). It would make more sense to me to re-run the algorithms in the setup used in the paper. Moreover, the algorithms have been run only for 3 seeds, which is far too little since these are not image-based environments.


**Questions:**

Questions
1. I don’t fully understand why model-based helps here. Aren’t the issues of OOD actions still relevant given that the transition model is a function of the action and the action is sampled from the evaluation policy?
2. Is this method scalable to when multiple policies generate the fixed dataset? And can we have a behavior policy-agnostic version where we don’t know what policies generated the data?
3. While this paper is different, there does seem to be some relation to [1]. In that paper, they learn the state density ratio and use that for better control. Can the authors comment on the difference between how this paper uses the state-density ratio vs. how [1] uses the state-density ratio (not the ratio learning part)?

[1] Off-Policy Deep Reinforcement Learning by Bootstrapping the Covariate Shift. Gelada et al.


**Limitations:**

See above.

---

> ### Author Rebuttal · Authors · 2023-08-10
>
> We thank the reviewer for the insightful and detailed comments. We respond to specific questions and comments as bellow.
> ## 1. Concerns
> Concern 1: Table is hard to read
>
> In response to your suggestion, we have revised the paper and included two modified tables in the pdf attached. In these tables, we have highlighted the scores that are larger than 90% of the largest score.
>
> Concern 2 : Why state-based method helps in our method？
>
> To clarify the contribution of our state-based method, we compare it with CQL and COMBO.
>
> As shown below, CQL underestimates state values point-wisely, while COMBO and CSVE underestimate state values on expectation.
> - CQL:
> $$ \mathbb E_{\pi(a|s)}[\hat{Q}^{\pi}(s, a)] \leq \mathbb E_{\pi(a|s)}[Q^{\pi}(s, a)], \forall s \in D $$
> $$ \hat{V}^{\pi}(s) \leq V^\pi(s) , \forall s \in D $$
>
> - COMBO:
> $$\mathbb E_{s \sim \mu_0, a \sim \pi(a|s)}[\hat{Q}^{\pi}(s, a)] \leq \mathbb E_{s \sim \mu_0, a \sim \pi(a|s)}[Q^{\pi}(s, a)]$$
> $$\mathbb E_{s \sim \mu_0}[\hat{V}^\pi(s)] \leq \mathbb E_{s \sim \mu_0}[V^\pi(s)] $$
>
> - CSVE:
> $$ \mathbb E_{s \sim d}[\hat{V}^{\pi}(s)] \leq \mathbb E_{s \sim d}[V^{\pi}(s)]$$
>
> where $\mu_0$ in COMBO represents behavior policy, $\hat{Q}^{\pi}$ is the estimation of $Q^{\pi}$,  and $d(s)$ in CSVE represents any state distribution.
> The motivation for using the state-based method can be summarized as follows:
> - Compared to CQL, both CSVE and COMBO aim to achieve better performance by relaxing the conservative estimation guarantee from point-wise state values to the expectation of state values. However, their conservative approaches differ: CSVE directly penalizes out-of-distribution (OOD) states, while CQL and COMBO penalize OOD state-action pairs.
> - In comparison with COMBO, by directly penalizing the OOD states, CSVE obtains the same lower bounds but under a more general state distribution. This offers a more flexible space for algorithm design, which is one of the main reasons for penalizing $V$ rather than $Q$.
> - By controlling the distance of $d$ to the behavior policy's discounted state distribution $d_u$, CSVE has the potential for further performance improvement.
>
> The bound is provided in the comparison with prior work in Section 3.1. We will refine this comparison in the future version.
>
> Concern 3: Questions about evaluation
>
> We understand your concerns regarding the comparison of results in Table 1. It is worth mentioning that copying results from prior work is a common practice in the offline literature; for instance, PBRL copy-pasting results from TD3-BC. However, we recognize the importance of ensuring a fair comparison across different setups. We have attempted to reproduce some of the results from previous work, but most of our reproduced results were inferior to those reported in the original studies. For example, we used the source code of COMBO to reproduce their results (Appendix C.4), but we could not achieve the same level of stability as presented in their paper. Therefore, we opted to use the results reported in the original papers for a fair comparison.
>
> Recently, we have conducted additional experiments with seven more seeds on HalfCheetah-medium, HalfCheetah-medium-replay, and HalfCheetah-medium-expert tasks. The updated scores can be found in the attached PDF. The results are consistent with the results reported in the paper. Given the limit on time and available computing resources, we can not reproduce all experiments in this rebuttal window. In the future revision, we will re-run all experiments over 10 seeds and report the results.
>
> ## 2. Questions
>
> Q1: issues of OOD actions
>
> CQL learns the conservative Q-values of $Q(s, a)$ on transitions $\{(s, a, r, s')\}^N$ of dataset $D$ where the value overestimation can only be introduced by OOD actions on $s'$. We aruge that **this is also a limitation of CQL that hinders further performance improvement**, since it does not learn Q-values on any out-of-sample state $s \notin D$ even the state is in-distribution with respect to the behaviour policy.
>
>  In contrast to adding conservatisim on Q-values as CQL, **CSVE proposes the idea of imposing conservatism on state values and proves that this approach is better in theory.**
>
> The intuition is that proper exploration on out-of-sample but in-distribution states (i.e., the states near data as in our paper) has benefits on improving the learnt policy compared to the behaviour policy that collected the data. As a comparision, penalizing Q for OOD actions has no guarranttees on OOD states (in CQL), or require assumptions on a penalty coefficient(in COMBO). However, when directly penalizing V function, CSVE get the same lower bounds as COMBO but under more general state distribution (Section 3.1). This means for those OOD actions, we can still get lower bound its value by bounding the state value, thus decreasing the extrapolaation error. We will add this discussion in the updated version.
>
> Q2:  scalability to dataset generated by multiple policies
>
> Our method is indeed scalable to situations where the fixed dataset is generated by multiple policies. Throughout the development of our algorithm, we did not assume that we have access to the policy that generated the dataset. Moreover, in our experimental evaluation, the medium-expert dataset serves as an example of a mixed dataset, as it is generated by both medium-played policy and expert policy.
>
> Q3: Relation to [1]
>
> [1] aims to learn an accurate value function $V{\pi}$, whereas in our work, we focus on obtaining a lower bound for $E_{s \sim d}[V{\pi}(s)]$. In [1], it is necessary to learn the  state-density ratio to ensure a more precise estimation of $V{\pi}$. In contrast, our approach does not require learning this ratio. Lower bounding $E_{s \sim d}[V{\pi}(s)]$ allows us to avoid visiting out-of-distribution states when executing actions.
>
> We acknowledge that this paper is related to off-policy evaluation, and we will include it in the related work section of our updated version.

---

> > ### Comment · Reviewer_ze4E · 2023-08-11
> >
> > Thank you to the authors for their response. Few things:
> > - As for the table, I meant simply highlighting where your method does well (not every method that did better than some threshold)
> > - Sorry, but I dont fully understand why the issues of OOD actions do not arise. Ultimately, if CSVE is relying on a model, then that involves sampling actions from some policy different from the behavior policy, which means it would still suffer from OOD overestimation (since model was approximated using dataset). Moreover, to have a policy improvement algorithm, we do need $Q$, which involves sampling from $\pi$ (in equation 6), which can result in overestimation again. Is it fair to say: your method still suffers from OOD actions but only on states that appear close to the dataset as opposed to suffering from OOD actions on bad OOD states?
> >
> > Clarification on the last point would be appreciated. Thank you.

---

> > > ### Author Response · Authors · 2023-08-15
> > >
> > > Thank you for your valuable feedback.
> > >
> > > In response to the table highlighting concerns, we recognize that this issue has been raised by three reviewers. As such, we have strived to strike a balance among these suggestions to present our results more effectively.
> > >
> > > Regarding the issue of out-of-distribution (OOD) actions, we acknowledge that our method is not entirely immune to OOD action overestimation. However, the impact of OOD actions should be mitigated in our approach. On the one hand, during the value estimation in Eq.5, all states are present in the dataset, while the policy $\pi$ is constrained to be close to the behavior policy (first term of Eq.9), allowing for slight exploration with high confidence of non-overestimation ensured by the model prediction of next states. On the other hand, during the policy learning in Eq.9, the additional action exploration (second term of Eq.9) is strictly applied to states in the dataset and only provides a bonus to actions that (1) themselves and their model-predictive next-states are both close to the dataset (ensured by the model) and (2) their values are favorable even with conservatism.
> > >
> > > We will revise our statement on OOD actions to provide better clarity. We hope this explanation addresses your concerns, and we appreciate your suggestions for improvement.

---

> > > > ### Comment · Reviewer_ze4E · 2023-08-15
> > > >
> > > > Thank you to the authors for their response and for answering my question.
> > > >
> > > > It seems like the focus is on avoiding overestimation on states that are in the dataset/near the dataset, but its unclear how this translates to favorable performance on OOD states since all the training is done on in-distribution states.
> > > >
> > > > Is it fair to say: other works will suffer from overestimation on OOD actions AND OOD states, but yours will mitigate overestimation on OOD actions in OOD states BUT only in those OOD states close to the dataset? So in OOD states that are *not* near the dataset, your algorithm will suffer the same way as others, but the core improvement is only on OOD states *near* the dataset?

---

> > > > > ### Author Response · Authors · 2023-08-16
> > > > >
> > > > > Thank you for your feedback. It's important to note that the training is not solely conducted on in-distribution states. As evidenced by the second term of Eq. 5, the action is selected by $\pi$, which means we also minimize the value of out-of-support states ($s'$). In OOD states near the dataset, our method provides a bound for overestimation, as follows:
> > > > >
> > > > > $$ \mathbb E_{s \sim d}[\hat{V}{\pi}(s)] \leq \mathbb E_{s \sim d}[V{\pi}(s)]$$
> > > > >
> > > > > Here, $d$ represents the distribution of near-dataset states, and $\hat{V}{\pi}$ is the estimation. This is a notable advantage over previous works, which are unable to provide such a bound. For instance, COMBO can only bound the value of in-distribution states.
> > > > >
> > > > > Regarding out-of-support states, our method cannot provide a suitable bound for them. However, it's crucial to mention that we do not explore these OOD states when deriving the policy. As a result, the OOD states should not cause any detrimental effects when outputting the policy.

---

> > > > > > ### Comment · Reviewer_ze4E · 2023-08-16
> > > > > >
> > > > > > Thank you to the authors for clarifying this point and I will raise the score.

---

> > > > > > > ### Author Response · Authors · 2023-08-19
> > > > > > >
> > > > > > > Thanks! We sincerely appreciate the valuable suggestions and shall improve the paper accordingly.

---

### Official Review · Reviewer_QSVW · 2023-07-06

**Soundness:** 3 good
**Presentation:** 3 good
**Contribution:** 3 good
**Rating:** 6
**Confidence:** 3

**Summary:**

<< I have read the authors' rebuttal and had raised my score based on the discussion >>

The paper discusses challenges in Reinforcement Learning (RL), particularly in real applications where online learning from scratch is often risky and unfeasible. To address this, the authors introduce Conservative State Value Estimation (CSVE), a novel offline RL approach that deviates from traditional methods that estimate conservative values by penalizing the Q-function on Out-Of-Distribution (OOD) states or actions. Instead, CSVE penalizes the V-function directly on OOD states. The authors present theoretical evidence that CSVE provides tighter bounds on true state values than Conservative Q-Learning (CQL), with similar bounds as COMBO but under more general discounted state distributions. This allows for potentially more effective policy optimization within the data support. The primary contributions of the paper include the proposal and theoretical analysis of conservative state value estimation, the introduction of a practical actor-critic algorithm applying CSVE, and an experimental evaluation demonstrating superior performance of CSVE over prior methods based on conservative Q-value estimation in tasks of Gym and Adroit in D4RL benchmarks. The simplicity of the proposed changes augments their potential for practical adoption in the field.

**Strengths:**

Clarity: The manuscript is effectively composed, offering straightforward comprehension. However, there's room for improvement in the explanation of certain equations. By simplifying these complex components, the reader's cognitive load could be significantly reduced.

Technical Soundness: The theoretical foundations of the paper are solid. The authors offer convincing theoretical derivations to support the proposed approach, contributing to the technical soundness of the paper.

Originality: Overestimation of action values represents a recurring challenge in the offline reinforcement learning landscape. The authors innovatively address this issue by learning a conservative estimate of state values and penalizing OOD states, offering a potentially tighter bound to the actual state value function. In my view, this represents a novel contribution to the field.

Significance: Attaining a tighter bound on the value estimate can considerably enhance performance in offline RL problems. Additionally, the simplicity of the proposed approach augments its potential for widespread adoption and application.

**Weaknesses:**

W1: The experimental results primarily focus on a range of continuous control tasks, neglecting discrete action space problems. Furthermore, the decision to use only three seeds for performance comparison appears limited, especially considering the growing trend to use ten seeds to apply metrics such as the Interquartile Mean (IQM) [1]. This raises concerns about the statistical validity of the proposed approach.

W2: The observed performance gains from implementing CSVE do not appear consistently significant across different domains or discernible patterns. The sporadic nature of these gains may question the efficacy of the proposed approach in diverse applications.

[1] https://arxiv.org/abs/2108.13264

**Questions:**

Q1: Assume we express Q(s,a) as R(s,a) + V(s') and learn a world model for one-step transitions rather than predicting Q(s,a) directly. In this case, minimizing the CQL objective in equation 1 would implicitly penalize the state value of OOD states. Could the authors explain how this approach differs from the one proposed in this paper?

Q2: What implications would an inaccurate world model have, specifically when the predicted next state $\hat{s'}$ doesn't match the actual next state ${s'}$?

Q3. Could the authors provide experiment results with more seeds?

**Limitations:**

The authors did not discuss the limitations of their work. No discussion needed regarding potential negative societal impact.

---

> ### Author Rebuttal · Authors · 2023-08-10
>
> W1: neglecting discrete action space problems and only three seeds for performance appears limited
>
> Recently, we have conducted additional experiments with seven more seeds on HalfCheetah-medium, HalfCheetah-medium-replay, and HalfCheetah-medium-expert tasks. The updated scores can be found in the attached PDF. The results are consistent with the results reported in the paper. Given the limit on time and available computing resources, we can not reproduce all experiments in this rebuttal window. In the future revision, we will re-run all experiments over 10 seeds and report the results.
>
> Though discrete action is excluded in our analysis, In future work, we aim to include a wider range of tasks to better demonstrate the effectiveness and versatility of the proposed approach.
>
>
>
> W2: The observed performance gains from implementing CSVE do not appear consistently significant .
>
> We have updated Table 1 and Table 2 in the attached PDF for a better illustration of the performance gains of our method. Considering the average score, our method is on par with PBRL in the gym domain and better than PBRL in the adroit domain. We agree with the reviewer that our method is not consistently significantly better than other methods, but overall it outperforms them.
>
>
>
>
>
>
> Q1: Minimizing the CQL objective in equation 1 would implicitly penalize the state value of OOD states
>
> When we express Q(s,a) as R(s,a) + V(s') and  minimize the CQL objective in equation 1 would indeed penalize the state value of OOD states. However, CQL underestimates state values point-wisely, while  CSVE underestimate state values on expectation.
> - CQL:
> $$ \mathbb E_{\pi(a|s)}[\hat{Q}^{\pi}(s, a)] \leq \mathbb E_{\pi(a|s)}[Q^{\pi}(s, a)], \forall s \in D $$
> $$ \hat{V}^{\pi}(s) \leq V^\pi(s) , \forall s \in D $$
>
>
> - CSVE:
> $$ \mathbb E_{s \sim d}[\hat{V}^{\pi}(s)] \leq \mathbb E_{s \sim d}[V^{\pi}(s)]$$
>
> CQL learns the conservative Q-values of $Q(s, a)$ on transitions $\{(s, a, r, s')\}^N$ of dataset $D$ where the value overestimation can only be introduced by OOD actions on $s'$. We aruge that **this is also a limitation of CQL that hinders further performance improvement**, since it does not learn Q-values on any out-of-sample state $s \notin D$ even the state is in-distribution with respect to the behaviour policy.
>
> The intuition is that proper exploration on out-of-sample but in-distribution states (i.e., the states near data as in our paper) has benefits on improving the learnt policy compared to the behaviour policy that collected the data. As a comparision, penalizing Q for OOD actions has no guarranttees on OOD states (in CQL). However, when directly penalizing V function, CSVE get the same lower bounds as COMBO but under more general state distribution (Section 3.1). This means for those OOD actions, we can still get lower bound its value by bounding the state value, thus decreasing the extrapolaation error. We will refine our discussion of this comparison in the updated version.
>
>
> Q2: What implications would an inaccurate world model have
>
> With an inaccurate world model, the theoretical part of our method will not be affected, given that the whole deduction does not require a model. However, the optimization part of Equation 9 will be affected. An inaccurate dynamics model may guide the policy to produce erroneous actions that result in high imaginary states.
>
> In the experiments, we find that the effect of model bias on RL performance is subtle in the medium tasks. We include experiments regarding different model errors in Appendix B. We use the average L2 error on transition prediction as a surrogate for model biases. Specifically, for the HalfCheetah task, there is no observable impact of model errors on scores, while in the Hopper and Walker2D tasks, there is only a slight decrease in scores as the errors increase.
>
>
> Q3. Could the authors provide experiment results with more seeds?
>
> We provide results over 10 seeds of three tasks in the pdf attached.  The results are consistent with the results reported in the paper. We will re-run all experiments over 10 seeds and report the results.

---

> > ### Comment · Reviewer_QSVW · 2023-08-18
> >
> > Thank you for addressing my previous concerns and providing additional results on the Cheetah problem using more seeds. I believe expanding this approach by using multiple seeds for other problems will further enhance the robustness of the results. I have a few more questions based on the provided feedback:
> >
> > Q4: From the presented results, it appears that CSVE outperforms the baselines when the data comes from an expert source. However, its performance seems to diminish when the data comes from a random or medium-quality source. Could you discuss why the quality of the data source seems to have a more impact on CSVE's performance compared to other baselines?
> >
> > Q5: I would appreciate further clarification on how the policy is derived using the conservative value function. In particular, how is the Q-function learnt or extracted, and what significance does the "conservative" value estimate hold within the context of AWR?

---

> > > ### Author Response · Authors · 2023-08-20
> > >
> > > Thanks for the valuable suggestions! We shall test with more seeds on other tasks as well.
> > >
> > > ## follow-up on Q4
> > >
> > > Not exactly. In fact, for mujoco tasks, CSVE has more advantage on datasets from random to medium types, while on expert datasets all algorithms perform already well and CSVE performs only better slightly or in parallel with others. For adroit tasks: (1) on 'human' and 'cloned' datasets, all algorithms fail in 3/4 tasks, while CSVE performs significantly better in the other 1/4 tasks; (2) on 'expert' datasets, all algorithms can work reasonably, while CSVE performs better in some tasks and in parallel with any baseline in other tasks.
> > >
> > > Thus, we think data quality has a significant impact on the absolute performance of all algorithms, and compared to baselines CSVE indeed has more advantage on datasets from random to medium types than expert datasets.
> > >
> > >
> > > ## follow-up on Q5
> > >
> > > The policy is derived by solving Eq.9, which balances the in-sample learning (the first term) and exploration based on conservative value estimation (the second term).
> > > - The Q function is learnt via Eq.6 during the value estimation phase. Since the AWR (the first term $L_{\pi}$ in Eq.9, defined in section 4.2) procedure is carried out only on $(s,a)$ pairs in the dataset, the significance of 'conservative' value estimation hold automatically.
> > > - Besides, the exploration term is carefully assured safe via the 'conservative' value estimation on predicative next-states.

---

> > > > ### Comment · Reviewer_QSVW · 2023-08-20
> > > >
> > > > Thank you for addressing my concerns. I have raised my score towards acceptance. I hope to see results on more seeds in the final manuscript in case the paper gets accepted.

---

> > > > > ### Author Response · Authors · 2023-08-21
> > > > >
> > > > > Thanks! We sincerely appreciate the valuable discussions, and shall update the paper accordingly.

---

### Official Review · Reviewer_EMsr · 2023-07-07

**Soundness:** 3 good
**Presentation:** 2 fair
**Contribution:** 3 good
**Rating:** 6
**Confidence:** 3

**Summary:**

In this paper, the authors propose an offline RL algorithm for conservative state value estimation (CSVE). This work is different than prior works that learn conservative state-action values like CQL or COMBO, in that it penalizes OOD states rather than OOD state-actions.

The authors show that CSVE gets similar theoretical lower bounds to COMBO, but under more general state distributions. The practical version of the algorithm is similar to COMBO with the following key differences:

*  CSVE learns a NN value function and penalizes OOD states (rather than state-actions for the Q function)
*  the policy is updated using AWR rather than SAC
*  the learned dynamics models are used for 1-step rollouts to generate fictitious state transitions for the OOD state penalty
*  the learned dynamics model is additionally used to allow the policy to explore local 1-step transitions around the data

The authors evaluate the CSVE method on the Mujoco and Adroit tasks from the D4RL benchmark. Generally, they find that their method outperforms or matches the relevant baselines (CQL, COMBO, AWAC) on these tasks.

**Strengths:**

To the best of my knowledge, the proposed CSVE algorithm is novel and is an interesting alternative to COMBO. The authors present good theoretical and empirical results illustrating the potential benefits of CSVE over COMBO and CQL. Thus, I believe that further exploration of whether incorporating the conservative penalty into a state-action value Q function or the state value function in a wider range of settings is an interesting direction for future research. Thus, I believe this work provides a worthwhile contribution to the research field.

**Weaknesses:**

There are a lot of small syntactical and word choice errors that do distract the reader and should be addressed in future versions, but generally the ideas presented in the paper are still comprehensible and well organized.

It is unclear to me exactly how the "Model-based Exploration on Near States" is performed and how Equation 9 is being optimized. Specifically, it is unclear to me whether the 2nd term in Equation 9 is being optimized with some variant of DPG (like TD3 or SAC) with the gradient being taken through the learned dynamics model, or by some variant of AWR like with the rest of the algorithm. In my opinion, this design choice is quite significant and should be fully explained in the main body of the paper. Additionally, the tradeoff between the 2 terms in Equation 9 seems important and I would appreciate the ablation study over the different values for $\lambda$ to be in the main body of the paper.

The Experiments section is a bit hard to follow, mostly because Tables 1 and 2 are hard to interpret without assistance. I think the clarity of the results would greatly be improved by bolding the top scores, and including aggregate scores like is done in many prior offline RL works. This would make it much easier to interpret how CSVE compares to priors algorithms.

Finally, the current ablation study in section 5.2 needs more analysis otherwise it doesn't seem to add much to the main body of the paper. I think an ablation with accompanying analysis on varying $\lambda$ and $\tau$ would be more interesting to include in the main body of the paper as those parameters seem in my opinion to be more related to the novel components presented in this work.

**Questions:**

Are there any environments or data regimes where you expect to see a bigger improvement for CSVE relative to prior methods? The current results are a bit underwhelming considering that many of the D4RL tasks are pretty saturated. Perhaps evaluating on the half-cheetah jump or ant angle tasks that have been tackled in other offline MB RL approaches like COMBO could lead to a stronger result. Considering that you only require 1-step predictions from your dynamics model, you could also potentially test on environments with more complicated observations like images.

**Limitations:**

No obvious limitation.

---

> ### Author Rebuttal · Authors · 2023-08-10
>
> We thank the reviewer for the insightful and detailed comments. We are glad to hear that Reviewer EMsr believes that our work presents good theoretical and empirical results illustrating the potential benefits of CSVE.  We respond to specific questions and comments below.
>
> ## 1. Main concerns
>
> Concern 1: A lot of small syntactical error and word choice errors
>
> Thank you for pointing that out. We appreciate your feedback on the readability of our paper and recognize that there are some syntactical and word choice errors that might be distracting. In future revisions, we will give more attention to these issues, ensuring that the text is polished and easier to read.
>
> Concern 2: how the "Model-based Exploration on Near States" is performed
>
> The optimization of the second term in Equation 9 involves calculating the gradient through the learned dynamics model. This is achieved by employing analytic gradients through the learned dynamics to maximize the value estimates. It is important to note that the value estimates rely on the reward and value predictions, which are dependent on the imagined states and actions. As all these steps are implemented using neural networks, the gradient is analytically computed using stochastic back-propagation, a concept inspired by Dreamer[1]. Additionally, the detailed implementation can be found in the accompanying code provided with our paper.
> We agreed that this discussion is important and will include this in our updated version.
>
> Concern 3: The Experiments section is a bit hard to follow
>
>
> In response to your suggestion, we have revised the paper and included two updated tables in the attached PDF. In these tables, we have highlighted the scores that exceed 90% of the highest score. The average score is also provided for a more comprehensive comparison.
>
> Concern 4: More ablation should be added into the main body
>
> We've included an ablation study about different $\lambda$ in Appendix B. And according to your follow-up suggestion, we will place the ablation of $\lambda$ in our main text and move the current ablation study of $\beta$ to the appendix.
>
>
> ## 2. Questions
>
> > Q1: Any environments or data regimes that we expect to see a bigger improvement?
>
> We fully agree that it is valuable for further exploration of more applications. We check the halfcheetah-jump and ant-angle used in MOPO and COMBO. However, we run into some reproduction difficulty using the source code of COMBO（Appendix C.4). In future work, we would indeed love to test CSVE on tasks like Half-Cheetah Jump, Ant Angle, and environments with more complex observations such as images. This will allow us to better assess the method's potential and capabilities across a broader range of applications and compare it with existing approaches.
>
> In general, CSVE have advantage in the broader set of scenarios where the environment is well-defined real-world physical system and the behaviour policy is not optimal. Compared to IQL and CQL, CSVE has less value underestimation and better exploration on state-action pairs near the dataset.
>
> References:
> [1]Dream to control: Learning behaviors by latent imagination, ICLR 2020.

---

> > ### Comment · Reviewer_EMsr · 2023-08-18
> >
> > Thank you to the authors for their response. If they indeed follow through with the changes mentioned, then I would raise my score.

---

> > > ### Author Response · Authors · 2023-08-19
> > >
> > > Thanks! We sincerely appreciate the valuable suggestions and shall improve the paper accordingly.

---

### Author Rebuttal · Authors · 2023-08-10

We have included two updated tables in the attached PDF. In these tables, we have highlighted the scores that exceed 90% of the highest score. The average score is also provided for a more comprehensive comparison.

We have conducted additional experiments with seven more seeds on HalfCheetah-medium, HalfCheetah-medium-replay, and HalfCheetah-medium-expert tasks. The updated scores can be found in the attached PDF. The results are consistent with the results reported in the paper. Given the limit on time and available computing resources, we can not reproduce all experiments in this rebuttal window. In the future revision, we will re-run all experiments over 10 seeds and report the results.

---

### Decision · Program_Chairs · 2023-09-21

**Decision:**

Accept (poster)

**Comment:**

After discussion with reviewers, I am recommending acceptance of this work.

I would suggest the authors incorporate the feedback and suggestions provided by the reviewers in preparation of their camera-ready version.